# Efficient protein targeting to the inner nuclear membrane requires Atlastin-dependent maintenance of ER topology

Sumit Pawar[1], Rosemarie Ungricht[1], Peter Tiefenboeck[2], Jean-Christophe Leroux[2], Ulrike Kutay[1]*

[1]Department of Biology, Institute of Biochemistry, ETH Zurich, Zurich, Switzerland; [2]Department of Chemistry and Applied Biosciences, Institute of Pharmaceutical Sciences, ETH Zurich, Zurich, Switzerland

**Abstract** Newly synthesized membrane proteins are targeted to the inner nuclear membrane (INM) by diffusion within the membrane system of the endoplasmic reticulum (ER), translocation through nuclear pore complexes (NPCs) and retention on nuclear partners. Using a visual in vitro assay we previously showed that efficient protein targeting to the INM depends on nucleotide hydrolysis. We now reveal that INM targeting is GTP-dependent. Exploiting in vitro reconstitution and in vivo analysis of INM targeting, we establish that Atlastins, membrane-bound GTPases of the ER, sustain the efficient targeting of proteins to the INM by their continued activity in preserving ER topology. When ER topology is altered, the long-range diffusional exchange of proteins in the ER network and targeting efficiency to the INM are diminished. Highlighting the general importance of proper ER topology, we show that Atlastins also influence NPC biogenesis and timely exit of secretory cargo from the ER.
DOI: https://doi.org/10.7554/eLife.28202.001

*For correspondence:
ulrike.kutay@bc.biol.ethz.ch

Competing interests: The authors declare that no competing interests exist.

## Introduction

A characteristic feature of eukaryotic cells is the separation of reaction spaces into compartments that are delimited by membranes. The most elaborate cellular membrane system forms the boundary of a complex organelle - the endoplasmic reticulum (ER). The ER serves as a central hub for protein entry into the secretory pathway and plays a vital role in lipid synthesis and calcium homeostasis. Although the ER is a continuous membrane system that is spread throughout the cytoplasm, it possesses structurally distinct domains such as the mostly perinuclear, ribosome-studded ER sheets or the dense reticular network of peripheral ER tubules (*Chen et al., 2013*; *Friedman and Voeltz, 2011*; *McNew et al., 2013*; *Shibata et al., 2006*; *Westrate et al., 2015*). The characteristic morphology of these ER domains is generated and maintained by proteins that partition into them; exemplified by the enrichment of the luminal ER spacing protein CLIMP63 in ER sheets and of the membrane curvature-inducing proteins of the reticulon and DP1/Yop1p families in ER tubules and at the edges of ER sheets (*Lin et al., 2012*; *Shibata et al., 2010*; *Voeltz et al., 2006*). The formation of the peripheral, tubular ER network relies on homotypic membrane fusion mediated by GTPases of the Atlastin family (*Hu et al., 2009*; *Moss et al., 2011*; *Orso et al., 2009*). Atlastins weave membranes together, establishing the polygonal ER that expands all the way towards the cell periphery (*Wang et al., 2016*). In vitro, Atlastins and membrane-tubule stabilizing factors are sufficient for the formation and maintenance of the tubular ER network (*Powers et al., 2017*). In living cells, ER network formation is further supported by microtubules and RabGTPase-dependent growth of new membrane tubules (*Audhya et al., 2007*; *English and Voeltz, 2013*; *Gerondopoulos et al., 2014*).

The boundary of the nuclear compartment - the nuclear envelope (NE) - is formed by a specialized ER membrane sheet that surrounds and protects chromatin (*Anderson and Hetzer, 2008*; *Ungricht and Kutay, 2015*). Reflecting functional specialization of the two NE lipid bilayers, the inner and outer nuclear membranes (INM and ONM) possess a distinct protein composition, even though both membranes are joined at numerous sites to form membranous pores. Compared to the ONM and the ER, the INM contains a unique set of membrane proteins that interact with nuclear lamins and chromatin, thereby playing a critical role in nuclear organization and genome regulation (*de Las Heras et al., 2013*; *Wong et al., 2014*).

Targeting of newly synthesized integral membrane proteins from the ER to the INM occurs within a system of continuous membranes. To reach the INM, membrane proteins must pass through nuclear pore complexes (NPCs), which control nucleo-cytoplasmic exchange. How newly synthesized integral membrane proteins traverse NPCs and are enriched at the INM has been a focus of intensive research, resulting in several models that explain protein targeting to the INM (*Katta et al., 2014*; *Ungricht and Kutay, 2015*; *Zuleger et al., 2011*). The initially proposed 'diffusion-retention model' posits that proteins diffuse from the ER to the INM, where they accumulate due to interactions with nuclear retention partners (*Powell and Burke, 1990*; *Soullam and Worman, 1995*). However, both the energy-dependent enrichment of a reporter protein at the INM in human cells (*Ohba et al., 2004*) and the description of transport receptor-mediated import of the yeast proteins Heh1 and 2 (*King et al., 2006*; *Meinema et al., 2011*) challenged this model, suggesting that INM targeting of the studied proteins may occur by an active process. Yet, recent kinetic analyses and mathematical modeling of protein sorting to the INM affirmed that diffusion and retention sufficiently explain the targeting of several INM proteins in human cells (*Boni et al., 2015*; *Ungricht et al., 2015*). Biochemical reconstitution further revealed that INM targeting of the investigated INM proteins is independent of the nucleo-cytoplasmic transport machinery (*Ungricht et al., 2015*). Still, depletion of nucleoside triphosphates (NTPs) affected the efficiency of INM protein targeting, which seems, at first glance, to contradict a diffusion-retention based mechanism. At the same time, however, energy depletion also induced changes in ER network topology and reduced the long-range diffusional exchange of proteins in the ER (*Ungricht et al., 2015*).

A causal connection between ER network topology and the energy requirement of INM targeting has so far not been established. Here, we show that INM protein targeting depends on GTP hydrolysis. We link this need for GTP hydrolysis to the maintenance of ER dynamics and connectivity, which in turn influence the apparent diffusional mobility of proteins within the ER network. Further, we demonstrate that the function of Atlastin GTPases in establishing and preserving ER structure is critical to sustain the efficient targeting of proteins to the INM. Our work explains why diffusion-retention-based protein sorting is energy-dependent and emphasizes the importance of ER network topology for cellular functionality.

## Results

### Efficient INM targeting is dependent on GTP hydrolysis

To study the molecular requirements and kinetics of protein targeting to the INM, we had previously established an in vitro system that uncouples integration of membrane proteins into the ER from their translocation to the INM (*Ungricht et al., 2015*). In this assay, initial trapping of GFP-tagged, INM-destined reporter proteins in the ER of HeLa cells is achieved by enlargement of their extralumenal, nucleoplasmic domains (ND) by two RFP moieties that can be cleaved off by TEV protease (*Figure 1A*). Addition of TEV protease to the semi-permeabilized cells allows for a fast cleavage of the tandem RFPs, thereby enabling NPC passage of the reporters. Targeting to the INM is then reconstituted by supplementing HeLa cell lysate and an energy regenerating system (ATP, GTP, creatine phosphate and creatine kinase; later referred to as energy) (*Ungricht et al., 2016*). Accumulation of the GFP-tagged reporter proteins at the NE over time can be followed by time-lapse microscopy allowing for a quantitative description of the INM targeting process. Note that NE accumulation of the reporter proteins majorly reflects INM localization (*Ungricht et al., 2015*; *Zuleger et al., 2011*).

Our previous analyses of transport kinetics and requirements of three different INM proteins, i.e. lamin B receptor (LBR), SUN2 and lamina-associated protein 2β (LAP2β), had demonstrated that

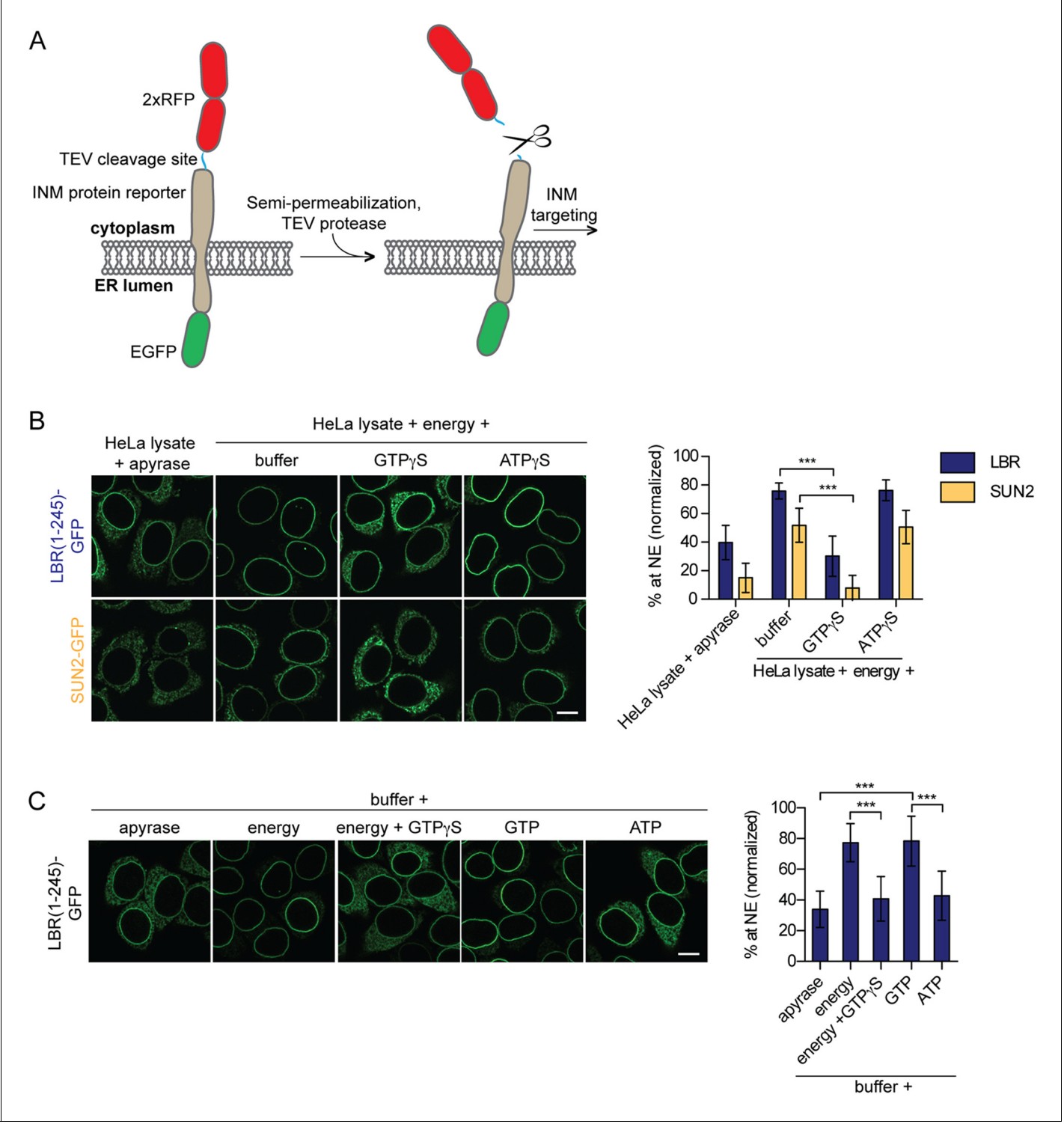

**Figure 1.** Targeting of membrane proteins to the INM in vitro is dependent on GTP hydrolysis. (**A**) Schematic representation of the in vitro transport assay for INM proteins (*Ungricht et al., 2015*). (**B**) Targeting of LBR(1-245)-GFP and SUN2-GFP was reconstituted in semi-permeabilized HeLa cells. INM targeting reactions were performed in presence of HeLa cell extract that was either depleted of NTPs by apyrase treatment or supplemented with an energy-regenerating system, and 0.3 mM GTPγS or ATPγS, as indicated. After incubation of cells at 37°C for 45 min, cells were fixed and imaged by confocal microscopy. NE enrichment was quantified as the ratio of the integrated fluorescence intensity at the NE to the total fluorescence intensity (in the ER and the NE), normalized relative to the fraction at the NE before release by TEV. Mean ±SD; n ≥ 63; ***p<0.0001. (**C**) Reconstitution of LBR(1-

*Figure 1 continued on next page*

Figure 1 continued

245)-GFP targeting for 45 min in the presence of buffer supplemented with either apyrase, buffer with an energy regenerating system in absence or presence of 0.3 mM GTPγS, or buffer supplemented with 0.75 mM GTP or ATP. Mean ±SD; n ≥ 139; ***p<0.0001. Scale bars: 10 µm.

DOI: https://doi.org/10.7554/eLife.28202.002

their efficient targeting to the INM depends on the presence of NTPs (*Ungricht et al., 2015*). Based on this data, we now aimed at identifying the cellular factors that utilize NTPs to promote INM protein targeting. We first investigated whether relocalization of LBR and SUN2-derived reporters from the ER to the NE can be challenged by addition of non-hydrolysable ATP or GTP analogs. Targeting of LBR(1-245)-GFP and SUN2-GFP was reconstituted in semi-permeabilized HeLa cells for 45 min, followed by fixation and analysis of targeting levels. As previously reported, in presence of HeLa cell lysate and energy, LBR and SUN2 accumulated at the NE efficiently, but to different levels (*Ungricht et al., 2015*). Strikingly, NE enrichment of both LBR and SUN2 was severely compromised upon addition of 0.3 mM GTPγS, comparable to the levels attained upon depletion of NTPs by apyrase (*Figure 1B*). A subtle defect was also observed with increasing concentrations of ATPγS (data not shown). This reduction was, however, weaker than that observed with GTPγS.

To test whether NTPs were sufficient for INM targeting, we performed reconstitution experiments exclusively in the presence of single nucleotides in buffer, that is, without addition of cell extract. Note that semi-permeabilization of cells results in a wash out of soluble cytosolic components. Remarkably, NE enrichment of LBR(1-245)-GFP could be fully reconstituted by the mere addition of GTP to semi-permeabilized cells, comparable to the levels obtained in presence of a complete energy regenerating system (*Figure 1C*). In contrast, addition of ATP did not suffice to support NE accumulation of LBR(1-245)-GFP beyond levels achieved by addition of buffer alone. Together, these results suggest that efficient enrichment of membrane proteins at the INM requires the enzymatic contribution of a GTPase present in the semi-permeabilized cells, indicating that the GTPase in question may reside in the ER or nucleus.

## GTPγS affects diffusion of membrane and soluble ER proteins

Exploiting kinetic analyses and mathematical modeling, we had previously attributed the energy requirement of INM targeting to the maintenance of ER organization and its influence on the diffusional exchange of proteins within the ER network (*Ungricht et al., 2015*). To analyze the effect of GTPγS on the diffusional properties of our INM targeting reporters in the ER, we measured their mobility by fluorescence recovery after photobleaching (FRAP). As we were interested in changes in long-range exchange of molecules, we performed FRAP on a relatively large area of the ER (~20 µm$^2$). Thus, the 'apparent' diffusion coefficients and mobile fractions that will be referred to in our study report on a combination of the diffusional properties of proteins in the 2D membrane plane and the influence of 3D geometry of the ER network on long-range exchange (*Sbalzarini et al., 2005*) (illustrated in *Figure 2A*). In line with the INM targeting defects, ER mobility of 2xRFP-tev-LBR (1-245)-GFP in semi-permeabilized cells supplemented with HeLa cell extract and energy was drastically compromised in the presence of GTPγS, but was unaffected by ATPγS (*Figure 2B*). Similar to INM-destined reporters, the diffusional mobility of the ER-resident membrane protein Sec61β-GFP was also strongly reduced upon treatment with 0.3 mM GTPγS, and was insensitive to the same concentration of ATPγS (*Figure 2C*).

Next, we assessed the diffusional properties of soluble ER lumenal proteins by fluorescence loss in photobleaching (FLIP) using GFP targeted to the ER lumen (GFP-KDEL) as a read-out in the same experimental set-up. A circular spot was repeatedly photobleached and the loss of fluorescence in the surrounding area as well as in a region on the opposite side of the nucleus was monitored. Rapid exchange of bleached fluorophores with unbleached molecules led to a simultaneous loss of fluorescence in the area surrounding the bleaching spot, but to a slightly delayed decrease in the distant region (*Figure 2D*, upper panels and graph). In the presence of GTPγS, loss of fluorescence in the surrounding area was significantly delayed and the fluorescence intensity on the opposite side of the nucleus even remained stable throughout the 12 min time course of the experiment (*Figure 2D*, lower panels and graph). Taken together, the observed defects on diffusional exchange of ER lumenal and membrane proteins suggest that GTPγS alters ER organization.

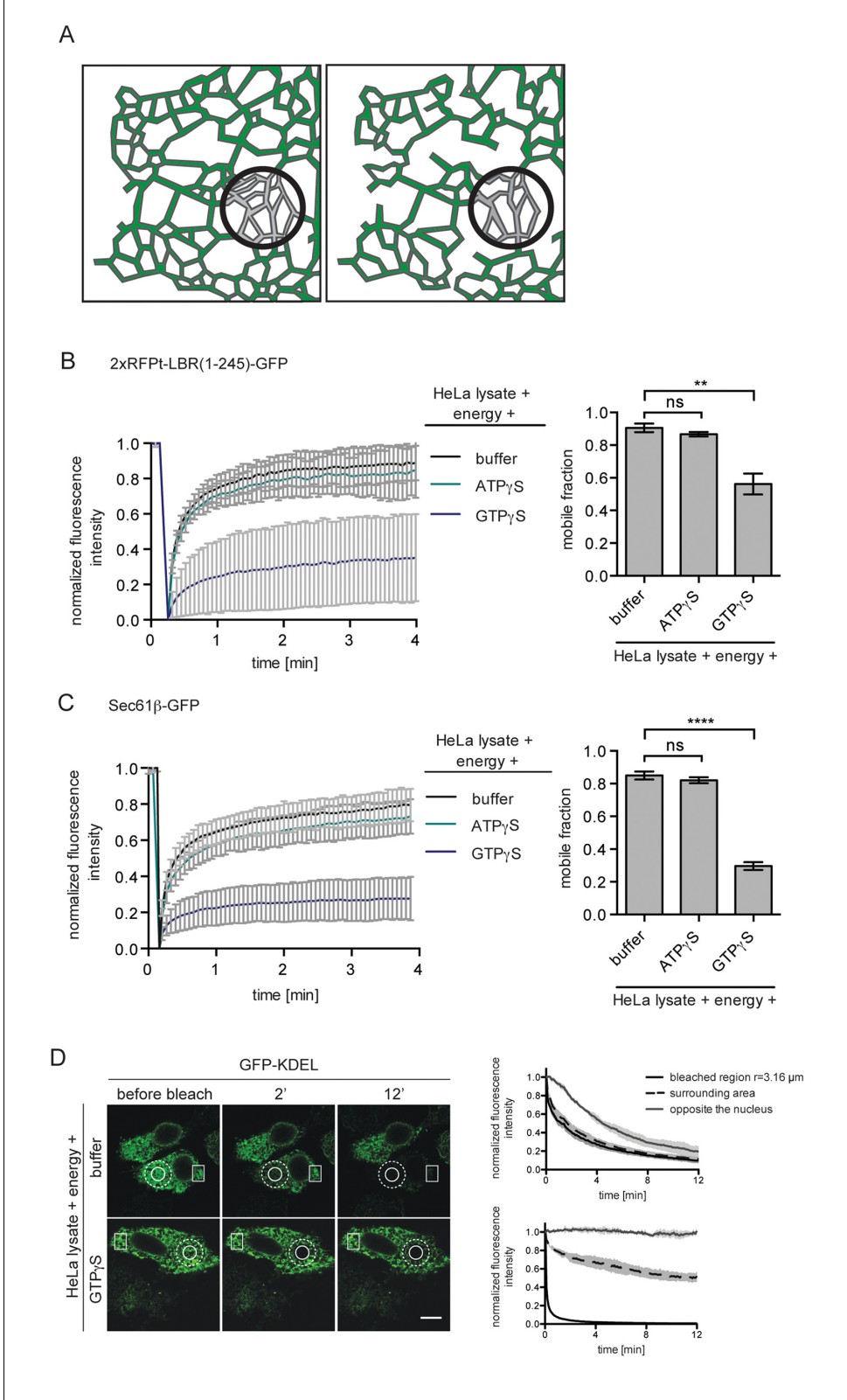

**Figure 2.** ER protein mobility is reduced in the presence of GTPγS. (**A**) Model illustrating the hypothetical differences in ER organization of cells with either a highly interconnected polygonal ER (left) or a more sparsely connected network of polygons and tubules (right). Upon bleaching the fluorescence of labeled proteins in the indicated circle, diffusion of fluorescent proteins from the surrounding ER network into the bleached area may occur with different kinetics due to the differences in ER organization. (**B**) FRAP on 2xRFP-tev-LBR(1-245)-GFP in the ER of semi-permeabilized cells in

*Figure 2 continued on next page*

*Figure 2 continued*

presence of HeLa cell lysate and energy, and further supplemented with buffer, 0.3 mM ATPγS or GTPγS. Recovery curves and mobile fractions derived from FRAP experiments. Mean ±SD (left curves); Mean ±SD (bar graphs); N ≥ 2; n ≥ 16; **p<0.01. (C) FRAP on Sec61β-GFP in the ER performed with semi-permeabilized cells as in (B). Recovery curves and mobile fractions derived from FRAP experiments. Mean ±SD (left curves); Mean ±SEM (bar graphs); N ≥ 4; n ≥ 32; ****p<0.0001. (D) FLIP performed on semi-permeabilized cells expressing GFP-KDEL in presence of HeLa cell lysate and energy, in absence and presence of 0.3 mM GTPγS. Mean ±SD; N = 3; n ≥ 9. Scale bars, 10 μm.

DOI: https://doi.org/10.7554/eLife.28202.003

## Continuous GTP hydrolysis is essential to maintain ER dynamics and mobility of proteins within the ER network

Our data so far indicate that GTP hydrolysis is required to sustain the diffusional mobility of soluble and membrane proteins in the ER, thereby contributing to the efficient protein trafficking towards the NE. However, our in vitro reconstitution experiments begin with semi-permeabilization of cells; a procedure that might affect ER topology. Therefore, we introduced a recovery time of 30 min in presence of HeLa lysate and energy after semi-permeabilization. Only after this recovery phase, nucleotide analogs were added (*Figure 3A*). Mobility of the reporters in the ER was monitored by FRAP, starting at 15 min of recovery, when 2xRFP-tev-LBR(1-245)-GFP already exhibited a high mobility with a mobile fraction of ~84%. Strikingly, when GTPγS was added after 30 min, the mobile fraction rapidly dropped below 40% (*Figure 3B*), comparable to the reduction in reporter mobility observed without recovery phase (*Figure 2B*). Addition of ATPγS did not alter the mobility of the reporter initially. However, after about 40 min, the mobility of the reporter started to decrease. When combining GTPγS and ATPγS, the drop in mobility was comparable to the addition of GTPγS alone. Thus, GTP hydrolysis is not merely required to repair the ER network post semi-permeabilization, but is necessary to maintain the diffusional mobility of ER membrane proteins.

The reduced mobility of both soluble and membrane ER proteins in the presence of GTPγS proposes a continuous and stringent requirement for GTP hydrolysis to maintain ER organization. This prompted us to visualize the changes that occur in the ER upon acute injection of GTPγS in live U2OS cells expressing EGFP-KDEL as a marker. In cells injected with GTP as a control, the ER appeared as a dense network that underwent significant remodeling; tubular extensions attached to neighboring membranes, initiating the formation of new junctions, while some polygons collapsed by approximation of tubules (*Video 1*). In contrast, when GTPγS was injected, the dynamics of the network was severely compromised. The characteristic polygon structure of the peripheral ER began to disappear within a few minutes, giving rise to sparsely connected bright membrane spots, potentially ER vesicles, and elongated tubules (*Figure 3C and E*, *Video 2*).

To capture changes in ER dynamics, we recorded movies starting 5 min after microinjection and quantified the number of emerging tubules that successfully attached to an opposing membrane in a defined 100 μm² area of the ER. Attachment was considered 'successful', if the emerging tubule formed a new junction and stayed at the opposing membrane. We also observed events during which the connection was unstable, classified as 'unsuccessful' (*Figure 3F*). Cells injected with GTPγS displayed a drastically reduced number of events as compared to cells injected with GTP. Moreover, the number of successful attachments decreased from >80% in control cells to less than half in GTPγS injected cells (*Figure 3F*). The global reduction in ER dynamics (and thus the decrease in total number of events) might be attributed to dysfunction of several cellular machineries. The marked reduction in successful membrane attachment events, however, hints towards a defect in homotypic ER membrane fusion - a function attributable to the GTPases of the Atlastin family (*Hu et al., 2009*; *Moss et al., 2011*; *Orso et al., 2009*).

## INM targeting depends on Atlastins

We next set out to test whether Atlastins are required for the efficient targeting of membrane proteins to the INM. Atlastins initiate the homotypic fusion of opposing ER membranes by dimerization of their GTPase domains. Subsequent conformational changes within the dimerized Atlastins linked to their GTPase cycle allow for close approximation of the membranes, followed by fusion (*Moss et al., 2011*; *Orso et al., 2009*). The Atlastin (ATL) family comprises three members in human cells, of which ATL2 and 3 are expressed in HeLa cells (*Rismanchi et al., 2008*). We downregulated ATL2 and ATL3 by RNAi in cells expressing the LBR and SUN2 targeting reporters and analyzed their

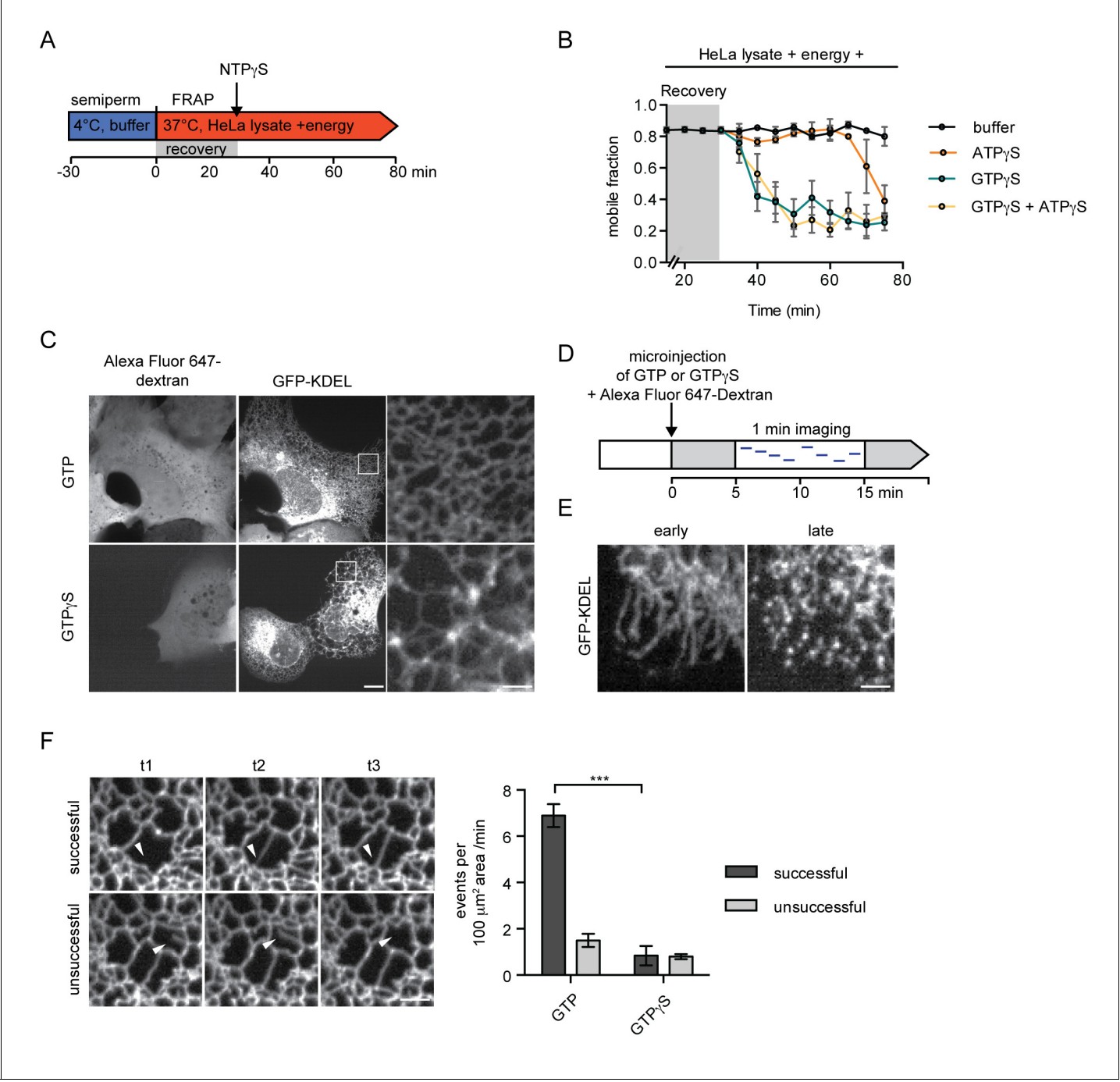

**Figure 3.** Inhibition of GTP hydrolysis affects ER morphology. (**A**) Schematic of experimental set-up in B. (**B**) 2xRFP-tev-LBR(1-245)-GFP expressing reporter cells were semi-permeabilized and allowed to recover in HeLa cell lysate and energy at 37°C. After 30 min, non-hydrolyzable ATP or GTP analogs were added to a final concentration of 0.3 mM. FRAP was performed every 5 min, starting during the recovery phase and continued after the addition of the nucleotide analogs. Mobile fractions over time. Mean ±SEM, N ≥ 3; n ≥ 20. (**C**) Morphology of the peripheral ER network of U2OS cells expressing GFP-KDEL after microinjection of either GTP or GTPγS. Alexa Fluor 647-labeled dextran served as injection marker. Scale bars, 10 μm, 2 μm for zoomed panels. (**D**) Schematic representation of the experiment in E and F. (**E**) U2OS cells expressing EGFP-KDEL were microinjected with a solution containing 10 mM GTP or GTPγS and a fluorescent dextran. 5 to 15 min after injection, cells were imaged over a time course of 1 min (1 frame/s) to visualize membrane dynamics in the peripheral ER network. Representative images of cells microinjected with GTPγS. Cells imaged early (after ~5 min) after microinjection displayed long unconnected tubules, cells imaged later (after ~15 min) showed fragmentation of the peripheral ER. Scale bar, 2 μm. (**F**) ER dynamics in the movies of (**E**) was quantified by scoring the number of successful (example in top panel) or unsuccessful membrane tubule

*Figure 3 continued on next page*

Figure 3 continued

attachments (example in bottom panel) in a 100 μm² area of cells injected with either GTP or GTPγS (*Videos 1* and *2*). Mean ±SEM; N ≥ 3; n ≥ 13; two 100 μm² squares for each cell; ***p<0.0005. Scale bar, 2 μm.

DOI: https://doi.org/10.7554/eLife.28202.004

accumulation at the NE using the in vitro system (*Figure 4*, *Figure 4—figure supplement 1*). Depletion of ATL2 alone led to a striking reduction in targeting of both LBR(1-245)-GFP and SUN2-GFP (*Figure 4*, *Figure 4—figure supplement 1* and *3*) and the expected changes in ER morphology, i.e. long, unbranched tubules in the peripheral ER (*Figure 4A*). Surprisingly, depletion of ATL3 using six different siRNA oligonucleotides (knockdown efficiency ≥80%) had no obvious effect on NE accumulation (*Figure 4*, *Figure 4—figure supplement 1C*) or ER morphology in HeLa cells (not shown).

To rule out compensation by remaining ATL3 after siRNA-mediated knock-down, we generated ATL3 knock-out (KO) cell lines using the clustered regularly interspaced short palindromic repeats (CRISPR)/Cas9 system. LBR reporter cells were transfected with the Cas9 nuclease and a single guide RNA targeting exon 2 of ATL3. Mutant isogenic clones were isolated, tested for protein expression and sequenced to confirm mutations in the target region (*Figure 4*, *Figure 4—figure supplement 2*). We did not observe obvious phenotypic changes in the ER in all three analyzed clones (*Figure 4A* and not shown). Knock-down of ATL2 in ATL3 KO cells, though, led to severe defects in ER morphology with even more long unbranched tubules and sparser connections as compared to the single depletion of ATL2 (*Figure 4A*). In agreement with the RNAi experiments (Figure S1), targeting of LBR(1-245)-GFP to the NE was efficient in ATL3 KO cells. In contrast, depletion of ATL2 strongly reduced NE accumulation of LBR(1-245)-GFP after 45 min (*Figure 4C*).

To assess whether Atlastin depletion influences the diffusional properties of membrane proteins in the ER, we analyzed the LBR reporter in semi-permeabilized cells by FRAP. In correlation with the INM targeting defect, ATL2 depletion by RNAi reduced the mobility of the reporter in the ER significantly, leading to a drop in both the size of the mobile fraction and apparent diffusion coefficient (*Figure 4D*). Knock-out of ATL3 alone had no effect. However, depletion of ATL2 in the ATL3 KO background further reduced the mobility of the membrane protein reporters to a similar extent as competition with GTPγS (*Figure 4D* and *Figure 2B*). We thus conclude that INM-destined proteins are compromised both in their ER mobility and INM targeting efficiency upon Atlastin depletion.

## Acute inhibition of Atlastin function is sufficient to cause INM targeting defects

As a second approach, we adopted a strategy to inhibit ATL function using truncated versions of human ATL2 and 3 that act as dominant-negatives on ER membrane fusion and morphology (*Hu et al., 2009*; *Wang et al., 2013*; *Wang et al., 2016*). These cytoplasmic Atlastin fragments (cytATL) dimerize with endogenous Atlastins via their GTPase domains, but block membrane fusion since they lack the necessary transmembrane segments. As expected, overexpression of cytATL2-RFP in U2OS cells led to the formation of long unbranched tubules and a significant reduction in the number of polygons in the peripheral ER compared to RFP alone or a cytATL mutant that abolishes dimerization (R244Q for cytATL2 and R213Q for cytATL3, (*Byrnes and Sondermann, 2011*)) (*Figure 5A*). In addition, the ER of cells overexpressing cytATL2-RFP appeared to be significantly less dynamic than the ER of cells overexpressing cytATL2(R244Q)-RFP or RFP alone (*Videos 3–5*). The number of successful attachments of ER tubules to neighboring membranes was reduced from 81% to 47% (*Figure 5B*), reminiscent of the reduction observed in cells injected with GTPγS (*Figure 3F*).

In order to perturb Atlastin function acutely during reconstitution of INM targeting in vitro,

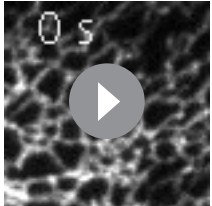

**Video 1.** ER remodeling in cells injected with GTP. ER dynamics in U2OS cells expressing GFP-KDEL microinjected with a solution containing 10 mM GTP and fluorescent dextran. Images were acquired with a spinning disk microscope at 1 s intervals for 1 min. The video is displayed at 4 frames/s. Image scale: 10 × 10 μm.

DOI: https://doi.org/10.7554/eLife.28202.005

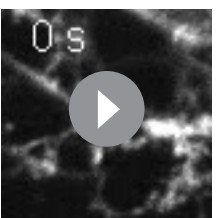

**Video 2.** Cells injected with GTPγS show drastically reduced remodeling events in the ER. ER dynamics in U2OS cells expressing GFP-KDEL shortly (>5 min) after microinjection with a solution containing 10 mM GTPγS and fluorescent dextran. Images were acquired with a spinning disk microscope at 1 s intervals for 1 min. The video is displayed at 4 frames/s. Image scale: 10 × 10 µm.
DOI: https://doi.org/10.7554/eLife.28202.006

we purified wild-type cytATL2 and 3 along with the dimerization-deficient RQ mutants from *E. coli*. Then, INM targeting of LBR(1-245)-GFP was reconstituted after pre-incubation of semi-permeabilized cells with the cytATL variants. Enrichment of LBR(1-245)-GFP at the NE was strongly impaired upon addition of wild-type cytATL2 or cytATL3, but was not affected by the addition of cytATL2(R244Q) or cytATL3(R213Q) (*Figure 5C*). The strong reduction in reporter accumulation at the NE upon interference with Atlastin function supports the notion that Atlastins are continuously required to preserve ER network integrity and demonstrates that these GTPases are needed to sustain efficient targeting of proteins to the INM. This requirement of Atlastins, in turn, could explain the observed GTP-dependent nature of INM targeting (*Figure 1*). It does, however, not exclude the contribution of additional GTPases.

A second class of GTPases implicated in structuring the ER network is the Rab GTPases (*Audhya et al., 2007*; *English and Voeltz, 2013*; *Gerondopoulos et al., 2014*). The function of Rabs in ER network formation in vitro can be counteracted by the Rab GDP dissociation inhibitor (RabGDI), which displaces Rabs from membranes (*English and Voeltz, 2013*; *Wu et al., 1996*). We used this approach to analyze whether the presence of the RabGDI affected INM targeting in our in vitro system. Even though Rab7 and Rab11 were efficiently extracted from membranes of semi-permeabilized cells with 20 µM recombinant GDI (*Figure 5*, *Figure 5—figure supplement 1A*), targeting of LBR(1-245)-GFP remained unaffected (*Figure 5*, *Figure 5—figure supplement 1B*). Also the combination of RabGDI and cytATL2 did not further decrease targeting efficiency, suggesting that Rabs are not involved in INM targeting or ER maintenance in the semi-permeabilized cell system.

## Interference with Atlastin functionality retards the kinetics of INM targeting in living cells

To validate the contribution of Atlastins to INM targeting in living cells, we adapted a system originally designed to study transport kinetics of proteins in the secretory pathway, termed Retention Using Selective Hooks (RUSH) (*Boncompain et al., 2012*). This system is based on overexpression of an ER resident protein as a fusion with core streptavidin (the hook), on which a reporter protein that is tagged with a streptavidin binding peptide (SBP) can be retained. Addition of biotin allows for a synchronous release of the reporter protein from the hook, for instance to study subsequent transport to the Golgi or the plasma membrane.

We exploited the RUSH system to trap INM-destined proteins in the ER by tagging them with SBP and using a fusion between core streptavidin and the ER protein STIM1-NN (a microtubule-binding deficient mutant of STIM1) as a hook (*Boncompain et al., 2012*). Then, we followed accumulation of the GFP-tagged reporter proteins at the NE upon their synchronous release from retention by biotin addition (*Figure 6A*). In agreement with the INM targeting kinetics that we had previously measured in semi-permeabilized cells (*Ungricht et al., 2015*), LBR(1-245)-SBP-GFP and LAP2β-SBP-GFP accumulated at the NE with different rates in living cells (*Figure 6B*). We confirmed that NE accumulation of the reporters reflected INM localization through antibody accessibility of their nucleoplasmic domains upon differential detergent permeabilization of the NE (*Figure 6*, *Figure 6—figure supplement 1*). To investigate how interference with Atlastin function in living cells affected INM targeting kinetics, we co-expressed the dominant-negative cytATL2-RFP fragment and, as negative controls, either cytATL2(R244Q)-RFP or RFP alone. Indeed, NE accumulation of both LBR(1-245)-SBP-GFP and LAP2β-SBP-GFP was delayed in cells overexpressing cytATL2-RFP ($t_{1/2}$ of LBR(1-245) increased from ~12 min to ~25 min), whereas cells overexpressing cytATL2(R244Q)-RFP or RFP as negative controls displayed unaltered targeting kinetics (*Figure 6C*). Note that we excluded cells from our analysis in which ER topology was severely compromised upon overexpression of cytATL2-RFP. In these cells, both LBR and LAP2β did not relocalize to the NE at all and stayed in the ER.

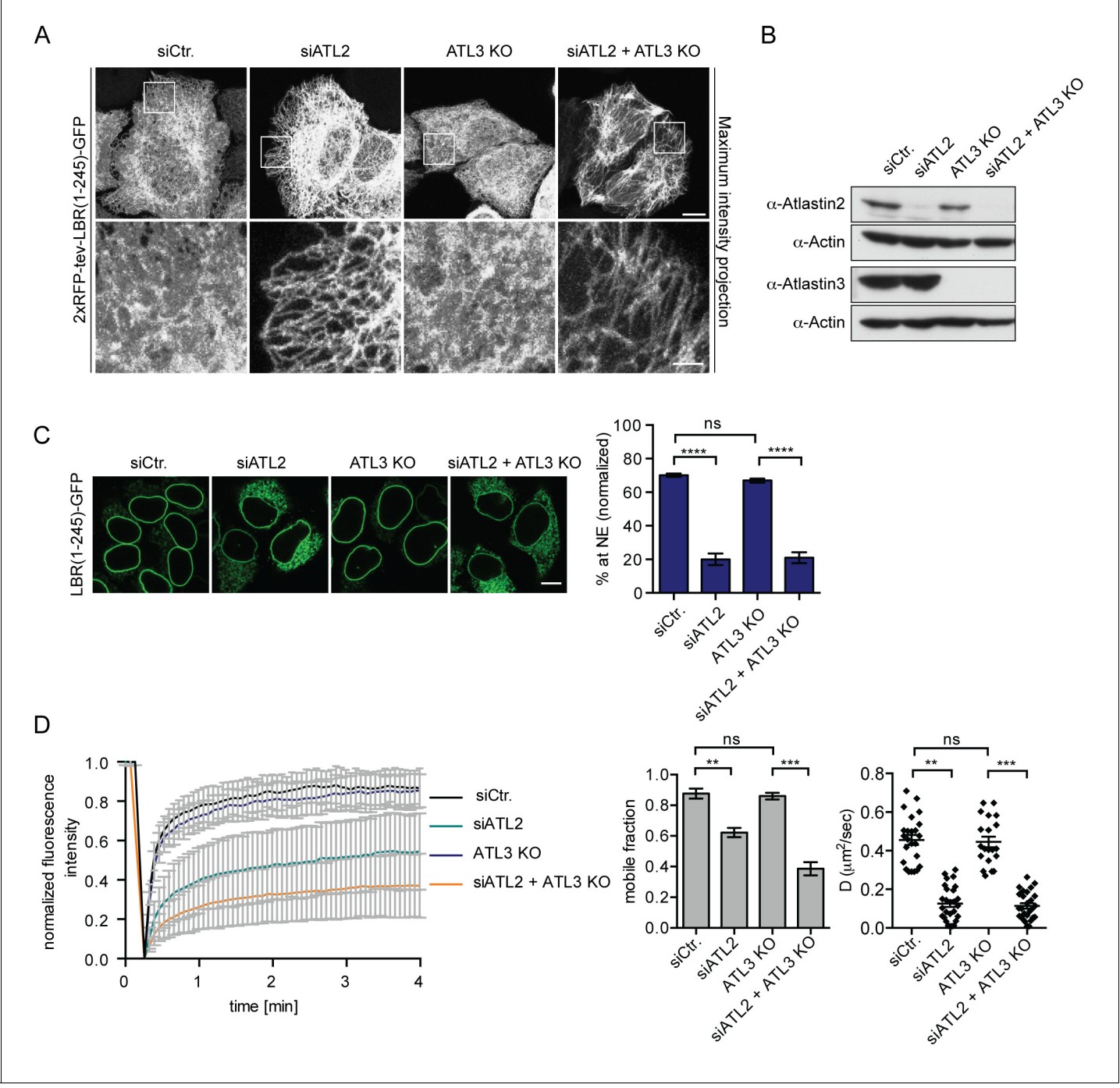

**Figure 4.** Atlastin 2 depletion impairs INM targeting in vitro. (**A**) Representative images of 2xRFP-tev-LBR(1-245)-GFP expressing reporter cells and ATL3 knockout (KO) reporter cells that were depleted for ATL2 by RNAi as indicated. Maximum intensity projection of 6 images taken at 0.37 µm steps from the bottom of the cell up to the equator of the nucleus. Scale bars: 10 µm (upper panels); 2 µm (lower panels). (**B**) Western blot confirming the depletion of ATL2 and the knockout of ATL3. (**C**) Accumulation of LBR(1-245)-GFP at the NE in a panel of control or Atlastin-depleted semi-permeabilized cells 45 min after TEV cleavage and addition of HeLa cell lysate and energy. Mean ±SEM; N = 4; n ≥ 239; ****p<0.0001. Scale bar, 10 µm. (**D**) FRAP in the ER of semi-permeabilized parental or ATL3 KO cells expressing the 2xRFP-tev-LBR(1-245)-GFP reporter. Cells were treated with control or ATL2 siRNAs as indicated and supplemented with HeLa cell lysate and energy after semi-permebilization. Mean ±SD (left graph). Mobile fractions and apparent diffusion coefficients derived from FRAP experiments; Mean ±SEM; N = 3; n ≥ 21; **p<0.005.

DOI: https://doi.org/10.7554/eLife.28202.007

The following figure supplements are available for figure 4:

**Figure supplement 1.** Targeting of LBR and SUN2 to the INM is ATL2-dependent.

*Figure 4 continued on next page*

*Figure 4 continued*

DOI: https://doi.org/10.7554/eLife.28202.008

**Figure supplement 2.** Validation of CRISPR/Cas9 knockout of ATL3 in HeLa cells.

DOI: https://doi.org/10.7554/eLife.28202.009

**Figure supplement 3.** NE accumulation of the LBR-derived reporter proteins in the Atlastin depletion experiment reflects INM targeting.

DOI: https://doi.org/10.7554/eLife.28202.010

Taken together, these data confirm the requirement of Atlastin GTPases for efficient targeting of INM proteins in living cells.

Alterations in ER morphology perturbed the efficient diffusional exchange of proteins in the ER network and affected the targeting kinetics of INM proteins. The kinetic delay in targeting of newly synthesized INM proteins induced by Atlastin depletion did not visibly affect the steady-state levels of endogenous LBR, LAP2β, SUN1, and emerin at the NE in the majority of unsynchronized interphase cells (*Figure 6*, *Figure 6—figure supplement 2A*), likely because newly synthesized INM proteins present only a minor fraction of the total pool at a given time. One process that is accompanied by the nearly synchronous enrichment of a larger pool of membrane proteins at the INM is NE reformation and growth after mitosis. Therefore, we also analyzed steady state localization of INM proteins in synchronized cells 12 hr after release from a thymidine block, thereby enriching the population for cells that just completed mitosis. These twin pairs of cells showed a visible accumulation of INM proteins in the ER, exemplified by immunofluorescence analyses of LBR, LAP2β, SUN1, and emerin upon co-depletion of ATL2 and ATL3 (*Figure 6*, *Figure 6—figure supplement 2B*). Notably, NPC biogenesis is known to critically depend on targeting of membrane proteins to the NE (*Antonin et al., 2005*; *Doucet et al., 2010*; *Talamas and Hetzer, 2011*; *Yavuz et al., 2010*). Indeed, immunostaining of nucleoporins revealed their striking mislocalization into annulate lamellae – stacks of ER sheets that contain NPCs (*Cordes et al., 1996*) – upon co-depletion of ATL2 and ATL3 (*Figure 6D*). This mislocalization was accompanied by a reduction in nucleoporin levels at the NE, indicative of a defect in chromatin-based NPC biogenesis. Taken together, these data provide evidence for the need of Atlastins to sustain efficient targeting of membrane proteins in living cells, which appears critical for the assembly of NPCs at the NE.

## Depletion of Atlastins influences the kinetics of ER to Golgi trafficking

We reasoned that ER network topology might also influence other cellular processes that depend on the efficient diffusion-based distribution of macromolecules in the ER network. One such process might be the delivery of cargo to ER exit sites for transport to the Golgi apparatus. To test whether changes in ER topology affect the kinetics of secretory trafficking from the ER to the Golgi, we used the RUSH system to study the trafficking of a well-characterized cargo protein – the vesicular stomatitis virus glycoprotein (VSVG) (*Mezzacasa and Helenius, 2002*). We expressed VSVG fused to SBP and GFP as a trafficking reporter and the invariant chain of MHC class II molecules (Ii) fused to core streptavidin as its hook (*Boncompain et al., 2012*) in HeLa cells (*Figure 7A*). Addition of biotin induced the rapid and synchronous transport of the VSVG reporter from the ER to the Golgi complex (*Figure 7*, *Figure 7—figure supplement 1*). Kinetic measurements of VSVG trafficking from the ER to the Golgi showed that VSVG rapidly enriched at bright puncta within 1 min in control cells (see bright puncta in siCtr., *Figure 7B*), consistent with VSVG accumulation at ER exit sites. VSVG then accumulated in the Golgi complex within 5 min. At 15 min, VSVG already started to be visible at the plasma membrane (*Figure 7B*). Strikingly, the enrichment of the reporter in the Golgi complex was delayed to a similar extent in cells depleted of ATL2 or both ATL2 and 3 but less so in ATL3-depleted cells. This retardation correlated with a delayed arrival of VSVG at ER exit sites (*Figure 7B*). Note that we only observed a kinetic delay in VSVG transport from the ER to the Golgi but not a block in trafficking, explaining why ER exit of a temperature-sensitive mutant of VSVG studied at longer time intervals seemed unaffected by depletion or deletion of Atlastins in previous studies (*Rismanchi et al., 2008*; *Zhao et al., 2016*). Together, these observations highlight the far-reaching consequences of aberrant ER morphology on cellular processes relying on the efficient exchange of proteins in the ER network.

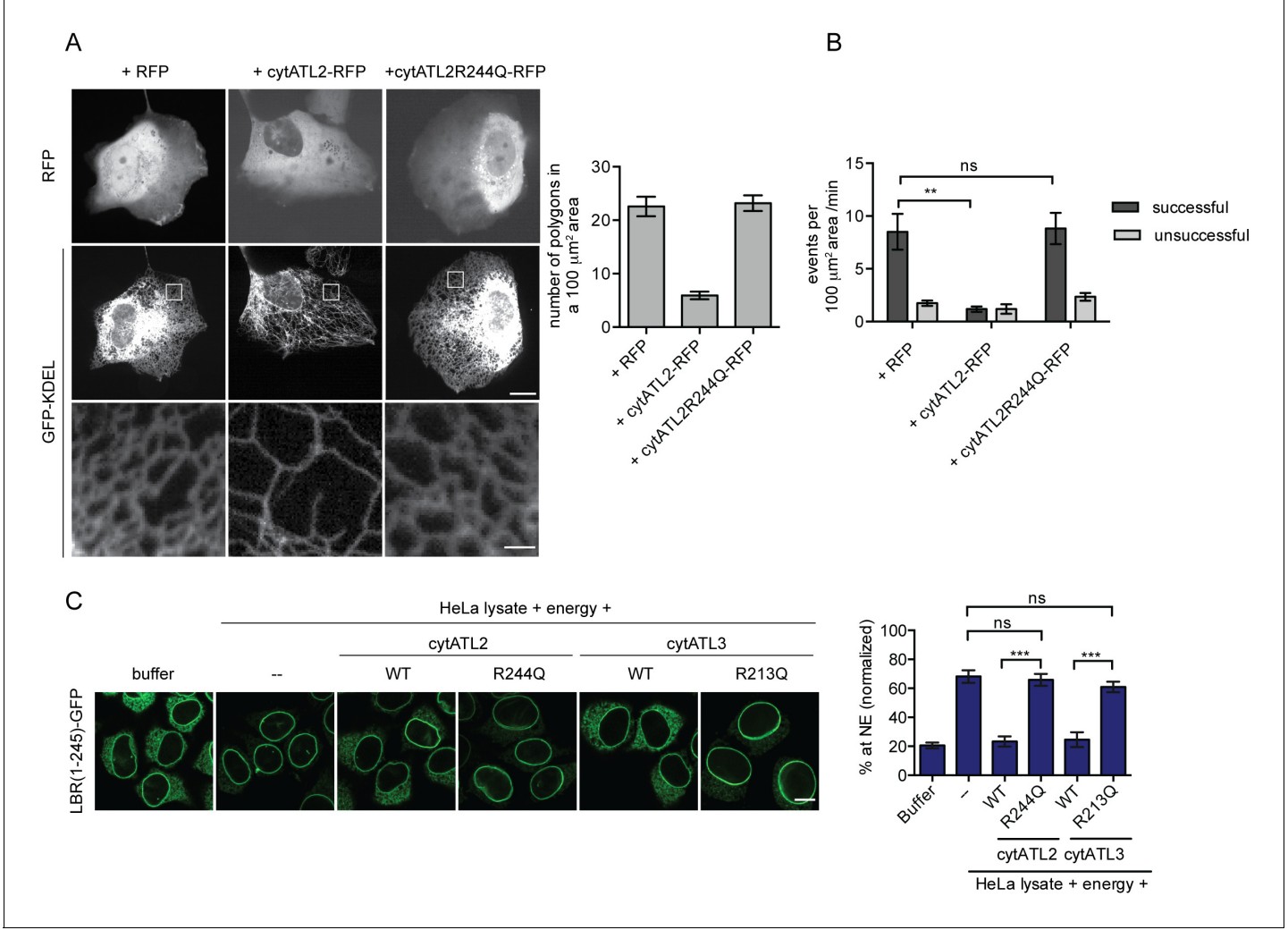

**Figure 5.** Dominant-negative Atlastins (cytATL) cause defects in ER structure, dynamics and INM targeting. (**A**) Representative images of the ER in U2OS cells overexpressing GFP-KDEL together with either RFP, cytATL2-RFP or cytATL2(R244Q)-RFP. The number of polygons within a 100 µm$^2$ area was quantified as a measure for ER connectivity. Mean ±SEM; N ≥ 3; n ≥ 13; one to two 100 µm$^2$ squares for each cell. Scale bars: 10 µm (upper panels); 2 µm (magnified lower panels). (**B**) ER dynamics was analyzed by live cell imaging in U2OS cells overexpressing RFP, cytATL2-RFP, or cytATL2 (R244Q)-RFP (*Videos 3*, *4* and *5*). The number of successful or unsuccessful membrane tubule attachments was quantified as in *Figure 3F*. Mean ±SEM; N = 4; n ≥ 16; two 100 µm$^2$ squares for each cell; \*\*p=0.005. (**C**) Targeting of LBR(1-245)-GFP to the NE in vitro in the presence of dominant-negative Atlastins. 2xRFP-tev-LBR(1-245)-GFP expressing reporter cells were semi-permeabilized. Then, the semi-permeabilized cells were pre-incubated with either buffer or 5 µM cytATL2, cytATL2(R244Q), cytATL3, cytATL3(R213Q) for 20 min at 37°C in buffer. After TEV cleavage of the reporter protein, energy and HeLa cell lysates supplemented with either buffer or 5 µM cytATL2, cytATL2(R244Q), cytATL3 or cytATL3(R213Q) were added. Reactions were allowed to proceed for 45 min, samples were fixed and NE targeting analyzed by confocal microscopy. Mean ±SEM; N ≥ 3; n ≥ 294; \*\*\*p<0.0005. Scale bar, 10 µm.

DOI: https://doi.org/10.7554/eLife.28202.011

The following figure supplement is available for figure 5:

**Figure supplement 1.** Addition of RabGDI does not impair INM targeting in vitro.

DOI: https://doi.org/10.7554/eLife.28202.012

## Discussion

In this study, we investigated the molecular requirements for trafficking of integral membrane proteins to the INM. Using non-hydrolysable nucleotide analogs, we demonstrate that INM targeting is GTP-dependent. GTP hydrolysis is essential to maintain ER network topology and dynamics (*Powers et al., 2017*; *Wang et al., 2016*), which, if perturbed, diminish the diffusional exchange of membrane proteins in the ER. In vitro and in vivo perturbation experiments link the observed

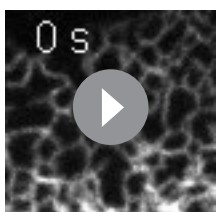

**Video 3.** ER undergoes extensive remodeling in control cells expressing RFP. ER dynamics in U2OS cells co-expressing GFP-KDEL and RFP. Images were acquired with a spinning disk microscope at 1 s intervals for 1 min. The video is displayed at 4 frames/s. Image scale: 10 × 10 µm.

DOI: https://doi.org/10.7554/eLife.28202.013

dependence of INM targeting on GTP hydrolysis to the functionality of Atlastins. Loss of Atlastin function mimics the effect of GTPγS on both ER topology and long-range diffusion within the ER, which in turn influences the kinetics of membrane protein delivery to the INM.

## GTP-dependence of INM targeting

Our reconstitution of INM targeting revealed that NE accumulation of the LBR and SUN2 reporter proteins was severely compromised in the presence of GTPγS. At first glance, this seems similar to the inhibition of nuclear import of soluble transport cargo in semi-permeabilized cells by non-hydrolysable GTP analogs (*Melchior et al., 1993*; *Moore and Blobel, 1993*; *Palacios et al., 1996*). Despite this analogy, however, the underlying mechanisms differ. Nuclear import of soluble cargo depends on the RanGTPase system that generates a high concentration of RanGTP in the nucleus. Nuclear RanGTP is needed for the dissociation of import substrates from shuttling nuclear transport receptors (*Görlich and Kutay, 1999*). If cytosolic Ran is bound to non-hydrolysable GTP analogs, the productive formation of importin-cargo complexes is perturbed. In contrast to import of soluble cargo, transport of SUN2 and LBR to the INM is not directly dependent on the RanGTPase system and the nucleo-cytoplasmic transport machinery for soluble cargo and not affected by addition of RanQ69L – a GTPase-deficient mutant of Ran that efficiently prevents importin-cargo interactions (*Ungricht et al., 2015*). Thus, the effect of GTPγS on INM targeting must have a distinct molecular cause.

Previous analysis had demonstrated that depletion of NTPs affects the diffusional mobility of INM-destined reporter proteins in the ER network (*Ungricht et al., 2015*). Similar to NTP depletion, we now observed that GTPγS leads to a severe reduction in the apparent diffusional mobility of membrane proteins in the ER. The reduction in diffusional exchange does not only concern INM-destined proteins, but also bona fide ER membrane proteins and even a soluble GFP within the ER lumen, consistent with changes in general ER topology induced by GTPγS.

Originally, it has been demonstrated that de novo ER network formation in vitro using *Xenopus* egg extracts requires GTP (*Dreier and Rapoport, 2000*). Recent observations revealed that an in vitro reconstituted ER network disassembles upon addition of GTPγS (*Powers et al., 2017*; *Wang et al., 2016*). Extending the latter finding, we demonstrate that GTP hydrolysis is needed to maintain ER network topology in semi-permeabilized and living human somatic cells. In striking correlation with these phenotypic defects, we measured a marked reduction in the mobility of membrane proteins in the ER of semi-permeabilized cells (*Figure 3B*) within a few minutes after addition

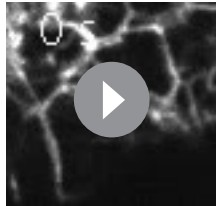

**Video 4.** ER dynamics in cells over-expressing dominant-negative ATL2 (cytATL2-RFP) is compromised, similar to cells injected with GTPγS (*Video 2*). ER dynamics in U2OS cells co-expressing GFP-KDEL and cytATL2-RFP. Images were acquired with a spinning disk microscope at 1 s intervals for 1 min. The video is displayed at 4 frames/s. Image scale: 10 × 10 µm.

DOI: https://doi.org/10.7554/eLife.28202.014

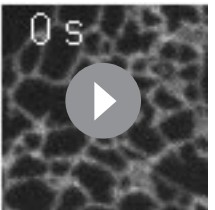

**Video 5.** ER remodeling is unaffected in cells expressing cytATL2(R244Q)-RFP. ER dynamics in U2OS cells co-expressing GFP-KDEL and cytATL2(R244Q)-RFP. Images were acquired with a spinning disk microscope at 1 s intervals for 1 min. The video is displayed at 4 frames/s. Image scale: 10 × 10 µm.

DOI: https://doi.org/10.7554/eLife.28202.015

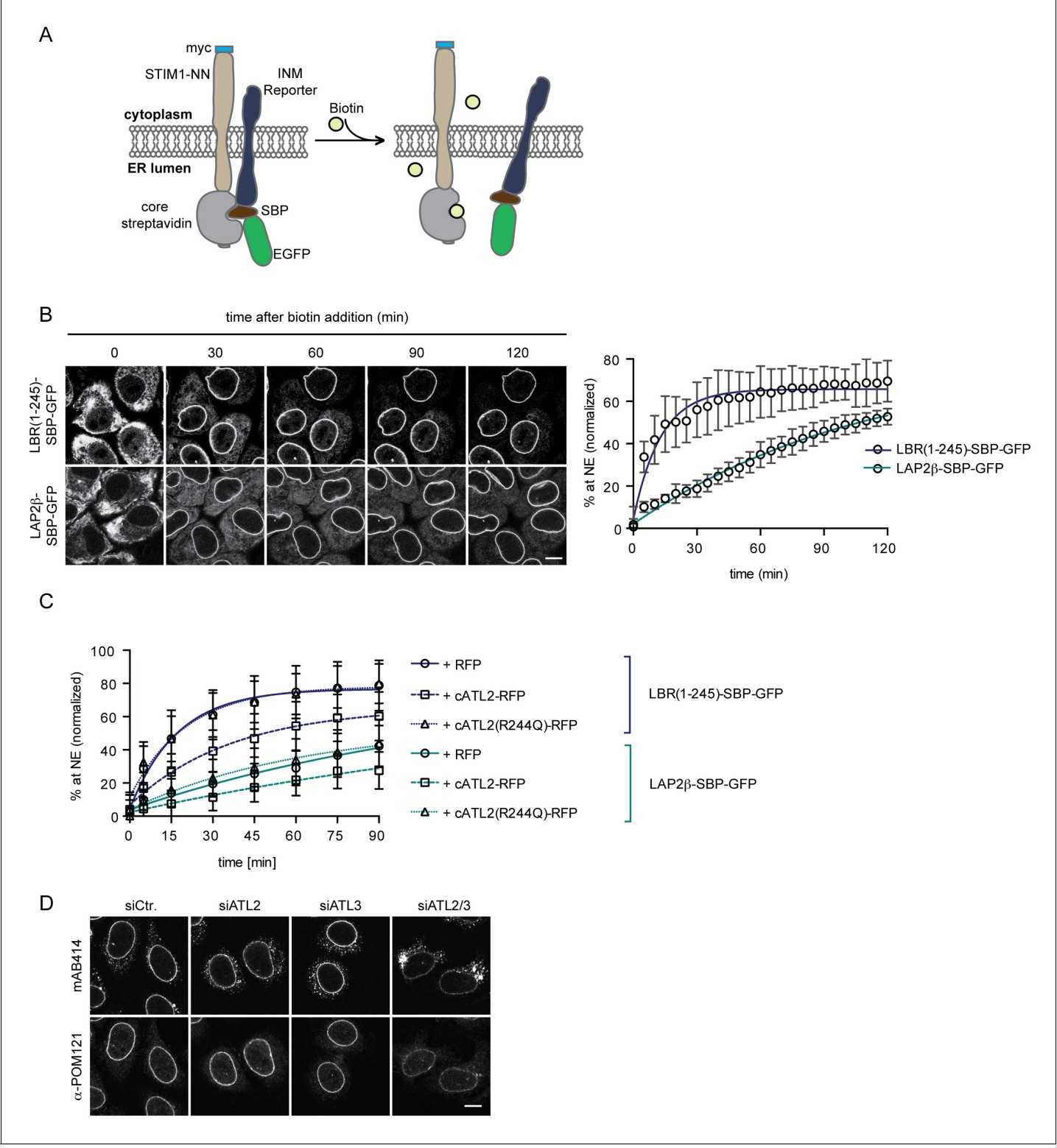

**Figure 6.** Loss of Atlastin function causes a kinetic delay in targeting of membrane proteins to the INM. (**A**) Cartoon representation of the Retention Using Selective Hooks (RUSH) system (**Boncompain et al., 2012**), adapted here for the analysis of INM targeting in living cells. (**B**) HeLa cells stably expressing STIM1-NN-Strep as hook and the LBR(1-245)-SBP-GFP or LAP2β-SBP-GFP INM targeting reporters were imaged by confocal live cell microscopy in 5 min intervals after release of the reporters from the hook by addition of 250 μM biotin. Accumulation of the reporters at the NE over time was measured using ImageJ. Mean ±SEM; N = 3; n > 76. Scale bar, 10 μm. (**C**) INM-RUSH reporter cells were transfected with cytATL2-RFP,

*Figure 6 continued on next page*

*Figure 6 continued*

cytATL2(R244Q)-RFP or RFP alone. After 24 hr, INM-destined reporters were released with biotin and their accumulation at the NE was followed by time-lapse microscopy and quantified as in B. Mean ±SD; N ≥ 2; n ≥ 24. (D) Immunofluorescence staining of cells depleted of Atlastins as indicated using mAB414 and POM121 antibodies. Note that co-depletion of ATL2 and ATL3 causes an NPC biogenesis defect, manifesting by reduced staining of the NE and accumulation of nucleoporins in cytoplasmic structures.

DOI: https://doi.org/10.7554/eLife.28202.016

The following figure supplements are available for figure 6:

**Figure supplement 1.** INM-RUSH reporters reach the INM after biotin-mediated release from the ER.

DOI: https://doi.org/10.7554/eLife.28202.017

**Figure supplement 2.** Localization of INM proteins after Atlastin depletion.

DOI: https://doi.org/10.7554/eLife.28202.018

of GTPγS. Thus, a requirement of GTP hydrolysis for the efficient targeting of proteins to the INM arises from its need for the generation and maintenance of ER network topology.

Atlastin GTPases have emerged as key players in the formation and maintenance of the tubular ER network (*Powers et al., 2017*). Similar to energy depletion and addition of GTPγS, depletion of Atlastins as well as treatment with dominant-negative Atlastin fragments impaired ER structure, membrane protein mobility in the ER and INM targeting, identifying an important role of the Atlastin GTPases and ER network topology for the efficient accumulation of membrane proteins at the INM. These observations also support the conclusions derived from our previous mathematical modeling approach, in which ER topology emerged as a key parameter influencing the diffusion-based distribution of membrane proteins in the ER network and thereby INM targeting kinetics (*Ungricht et al., 2015*). In an ER with reduced connectivity, as upon interference with Atlastin function, sorting of membrane proteins to the INM is delayed since proteins take longer to find their way to the INM. In the most extreme scenario of ER fragmentation, membrane proteins may even get stuck in disconnected parts of the ER, thereby leading to a reduced steady-state accumulation at the INM. Thus, GTP hydrolysis by Atlastins and ER network maintenance can explain why targeting of membrane proteins to the INM is energy-dependent.

Interestingly, we observed differences in phenotypic defects upon depletion of Atlastins. ATL2 depletion affected ER topology, membrane protein mobility and targeting to the INM, while the sole depletion of ATL3 did not. It is possible that ATL2 is a more potent fusogen than ATL3 as previously proposed (*Hu et al., 2015*) and/or that ATL3 is less abundant than ATL2 in HeLa cells. The aggravated defects in ER morphology that we observed upon co-depletion of ATL2 and ATL3 suggest that ATL3 function may become critical in the absence of ATL2. Whether Atlastins function redundantly and can compensate for the loss of each other, or whether the two Atlastins potentially mediate fusion at spatially distinct locations in the ER remains to be investigated. The potent impairment of INM targeting by cytATL3 may, however, suggest that Atlastins form mixed dimers under those conditions using excess recombinant protein, similar to what has been observed before (*Rismanchi et al., 2008*). Intriguingly, ER exit of VSVG was not only perturbed upon depletion of ATL2 but also slightly affected by loss of ATL3. As ATL3 depletion did not visibly affect the long-range diffusional exchange in the ER network, there might be a different cause for its influence on ER to Golgi transport. Although both Atlastins are known to be widely distributed within the ER network, there are notable differences in their localization. Compared to ATL2, ATL3 was found to be less aligned with microtubules (*Rismanchi et al., 2008*). It is thus conceivable that ATL3 depletion might more specifically affect the organization of ER membrane domains relevant for timely ER exit. Notably, all Atlastins have also been found associated with vesicular tubular clusters (VTCs) and the *cis*-Golgi. This may simply reflect ER escape of Atlastins accompanying the flux of membranes to the Golgi. Alternatively, it could also hint at a more specific role of Atlastins in maintaining the topology of the transitional ER.

Currently, we cannot exclude the involvement of additional GTPases in INM targeting. A contribution may for instance arise from other GTPases involved in structuring the ER network, like RabGTPases and the microtubule cytoskeleton (*English and Voeltz, 2013*). However, neither microtubule depolymerization (*Ungricht et al., 2015*) nor the extraction of Rabs did affect INM targeting in vitro (although not all Rabs may have been depleted with equal efficiency). Thus, if there is a contribution of these GTPases to INM targeting, it is not obvious in the in vitro system. Notably, Atlastin-

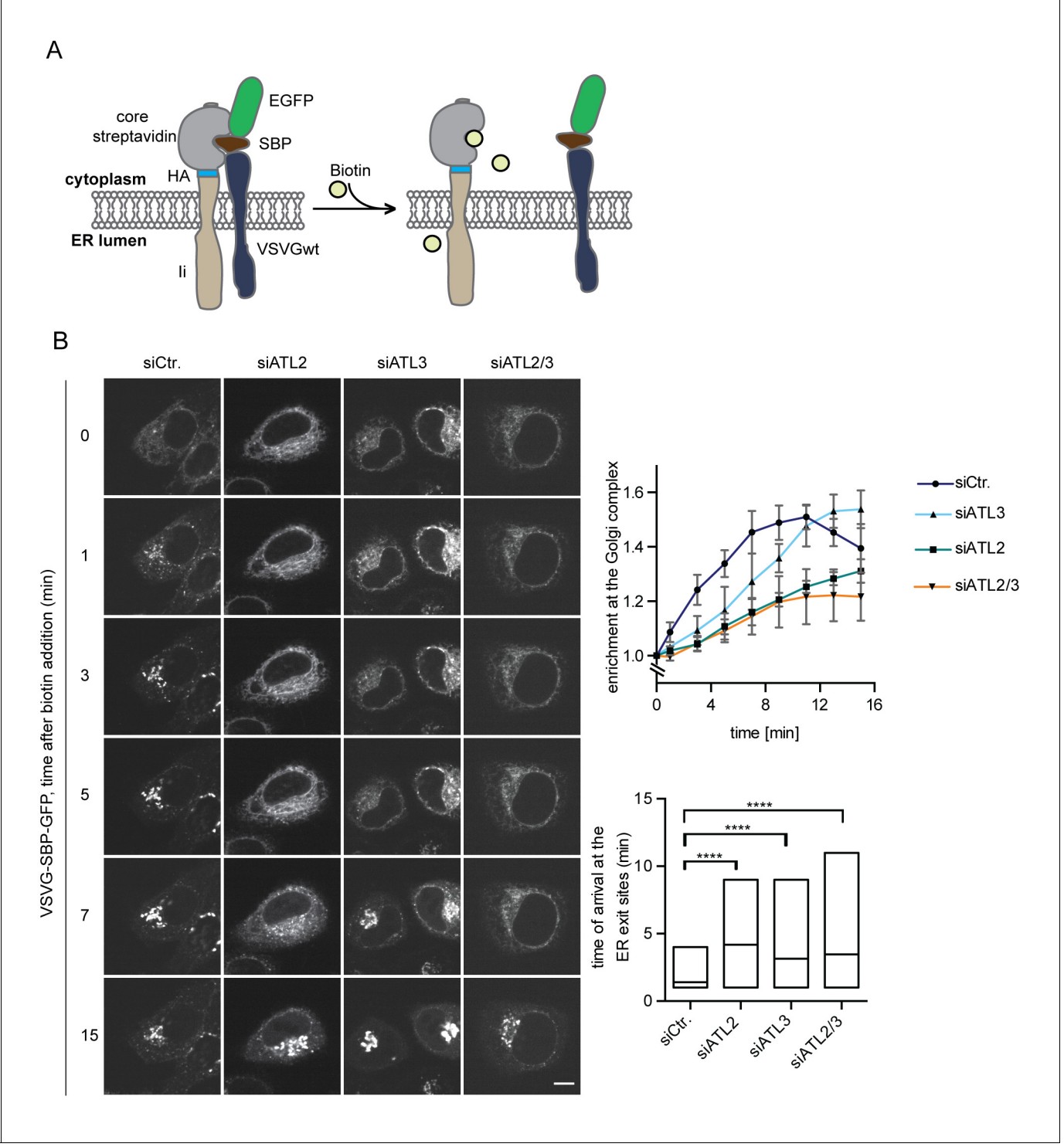

**Figure 7.** Atlastin depletion affects secretory trafficking. (**A**) Cartoon representation of the RUSH system to allow for synchronized secretory trafficking of membrane proteins out of the ER in live cells (*Boncompain et al., 2012*). (**B**) HeLa cells expressing Strep-Ii as hook and VSVG-SBP-GFP reporter were imaged by live cell microscopy in 1 min intervals after release of the reporters from the hook upon addition of 250 µM biotin. Enrichment of the reporter at the Golgi complex over time in control or Atlastin-depleted cells was measured using ImageJ (upper graph). Mean ±SEM; N = 3, n ≥ 21. Box plots representing the range of observed time points at which the reporter got enriched in bright puncta (lower graph). N = 3, n ≥ 21, ****p<0.0001. Scale bars, 10 µm.

DOI: https://doi.org/10.7554/eLife.28202.019

The following figure supplement is available for figure 7:

*Figure 7 continued on next page*

*Figure 7 continued*

**Figure supplement 1.** The VSVG(wt)-SBP-GFP reporter protein is transported to the Golgi complex after biotin-mediated release from the ER in Atlastin-depleted cells.

DOI: https://doi.org/10.7554/eLife.28202.020

mediated maintenance of the ER network is required early in the targeting pathway to the INM, upstream of other critical processes such as NPC translocation. Therefore, we cannot dismiss the possibility of other GTP-dependent steps in the targeting process. To address whether GTP hydrolysis by Altlastins is sufficient for INM targeting, we expressed Atlastin mutants with supposedly altered nucleotide specificity (from GTP to xanthosine 5' triphosphate, data not shown). However, the respective mutants behaved as dominant-negatives, hampering the analysis.

## Membrane connectivity and ER functionality

Despite its dynamic behavior and separation into structurally distinct domains, the continuity and interconnectivity of the ER network is preserved by frequent homotypic membrane fusion events. In general, in physiological settings, the ER is a continuous membrane system. Why is membrane continuity so important? For one, it may assure rapid flux of proteins and lipids from their site of synthesis to the site of function. Our study emphasizes that the continuity of the ER aids the timely delivery of proteins to the INM, thereby promoting the generation of a specific ER subdomain. A well-connected ER assures an unhindered diffusion of newly synthesized INM proteins to accompany the nuclear growth and NE expansion during interphase. The interconnected topology of the ER network is also important for the timely exit of newly synthesized cargo into the secretory pathway, as exemplified by the delayed trafficking of the model protein VSVG from the ER to the Golgi upon Atlastin depletion. Additionally, it may regulate interactions of the ER with several organelles by facilitating rapid exchange of proteins towards and away from contact sites.

Even during mitosis, the ER remains intact (*Ellenberg et al., 1997*), despite the large-scale reorganization of the NE-ER network. Interestingly, interference with Atlastin function early during in vitro nuclear assembly abrogates NE formation, whereas Atlastin inhibition at a later phase impairs NE expansion (*Wang et al., 2013*), indicating that ER connectivity is important for nuclear reformation after mitosis. We observed that a large fraction of NPCs is mislocalized to annulate lamellae upon depletion of Atlastins, which may reflect a function of Atlastins in post-mitotic NE reformation. During nuclear reassembly, INM proteins need to be recruited back to chromatin during a very short time window in late ana- and telophase, not only to support chromatin enclosure by membranes but also to promote post-mitotic NPC assembly (*Antonin et al., 2005*; *Mansfeld et al., 2006*). It is conceivable that this phase of the cell cycle is especially vulnerable to defects in ER network organization because the timely delivery of INM proteins from the ER to the reforming NE is crucial, leading to ectopic NPC assembly if perturbed. Alternatively, however, it is also possible that the altered connectivity in the ER may delay the recruitment of transmembrane proteins required for NPC formation from the ER to the NE during interphase (*Dawson et al., 2009*; *Doucet et al., 2010*; *Talamas and Hetzer, 2011*).

Finally, ER continuity and complexity seem of special importance in large and polarized cells like neurons. A group of inherited diseases, collectively termed hereditary spastic paraplegias (HSPs), arise due to mutations in ER shaping proteins and affect specifically neuronal cells with long axons (*Hübner and Kurth, 2014*). Neurons demand for an intricate ER topology, with a network of tubular projections and irregularly spaced cisternae reaching down all along the axon (*González and Couve, 2014*). The dimensions of the ER in neuronal cells might explain their vulnerability for mutations in ER shaping proteins and their demand for a potent fusogen as ATL1 (*Li et al., 2017*; *Rismanchi et al., 2008*; *Summerville et al., 2016*). Our study emphasizes how changes in ER network topology already influence trafficking of membrane proteins in small, non-neuronal cells. Such perturbations in ER morphology that cause subtle defects in protein trafficking are likely enhanced applied to neurons due to the long axonal extensions. We may thus propose that studying the diffusional status of membrane proteins in the ER and their trafficking towards destined organelles could help providing insights into mechanisms underlying ER morphology-associated diseases.

## Materials and methods

### Antibodies

The following commercial antibodies were used in this study: anti-ATL2 (rabbit; Abcam, ab 103084; RRID: AB_10711796), anti-ATL3 (rabbit; Proteintech, 16921–1-AP; RRID: AB_2290228), anti-β-actin (mouse; Sigma, A1978; RRID: AB_476692), mAB414 (mouse; Abcam, ab 24609; RRID: AB_448181), anti-LBR (rabbit; Abnova, PAB15583; RRID: AB_10696691), anti-LAP2β(mouse; BD transduction laboratories, 611000; RRID: AB_398313), anti-Lamin A/C (mouse; BD transduction laboratories, 612162; RRID: AB_399533), anti-Lamin B1 (rabbit; Abcam, ab 16048; RRID: AB_443298), anti-emerin (rabbit; Abcam, ab 40688; RRID: AB_2100059) and anti-Giantin (rabbit; Abcam, ab 24586; RRID: AB_448163). Antibodies against human POM121 (*Mansfeld et al., 2006*), and SUN1 (*Sosa et al., 2012*) have been previously described.

### Molecular cloning and generation of cell lines

All GFP constructs used in this study are enhanced GFPs (EGFP). HeLa cell lines carrying the tetracycline-inducible INM targeting reporters 2xRFP-tev-LBR(1-245)-GFP and 2xRFP-tev-SUN2-GFP have been described (*Ungricht et al., 2015*). For generation of stable HeLa cell lines expressing human Sec61β-GFP, a respective DNA fragment (*Ungricht et al., 2015*) was cloned into pIRESpuro2 vector (CMV promoter; Clontech).

The Myc-ATL2(pGW) vector was kindly provided by Craig Blackstone (NIH, Bethesda MD, USA). The human ATL3 ORF was obtained from the human ORFeome collection (14876, hORFeome V5.1). cDNAs encoding for cytoplasmic Atlastins fragments, ATL2(1–474) and ATL3(1–469), were cloned into the pQE30 vector (QIAGEN) for expression with an N-terminal His$_6$ tag in *E. coli*. For expression in mammalian cells, the cytoplasmic fragment of ATL2 was cloned into the pmRFP-N3 vector. Mutations in Atlastin constructs were generated using QuickChange (Agilent Technologies).

The vectors of the RUSH system components STIM1-NN_ManII-SBP-EGFP and Str-Ii_VSVGwt-SBP-EGFP (*Boncompain et al., 2012*) were kindly provided by Franck Perez (Institute Curie, Paris, France). To adapt the system to study INM targeting, stable cell lines expressing the hook and allowing for the tetracycline-induced expression of INM targeting reporters based on either LBR(1-245) or Lap2βwere generated. In brief, a DNA fragment encoding for myc-STIM1-NN-streptavidin was cloned into the pIRESpuro2 vector and randomly integrated into the genome of a HeLa cell line harboring an FLP recognition target (FRT) site and expressing the tetracycline repressor (TetR). The INM targeting reporters were generated by replacing ManII in the ManII-SBP-EGFP backbone by either LBR(1-245) or Lap2β. The respective DNA fragments encoding for LBR(1-245)-SBP-GFP or Lap2β-SBP-GFP were then subcloned into pcDNA5 FRT/TO (Invitrogen) for integration into the FRT site of the HeLa FRT/TetR cell line expressing the myc-STIM1-NN-streptavidin hook.

### Cell culture

HeLa FRT/TetR, HeLa K and U2OS cells were kind gifts from M. Beck (European Molecular Biology Laboratory, Heidelberg), D. Gerlich (Institute of Molecular Biotechnology, Vienna), and C. Azzalin (Instituto de Medicina Molecular, Lisboa) respectively. HeLa and U2OS cells were cultured in DMEM containing 10% FCS and 100 µg/ml penicillin/streptomycin, at 37°C and 5% CO$_2$. Expression of the 2xRFP-tev-LBR(1-245)-GFP and 2xRFP-tev-SUN2-GFP INM targeting reporters was induced with 20 ng/ml and 200 ng/ml tetracycline, respectively, 16 hr before the experiment. In cells depleted of Atlastins, 33 ng/ml and 333 ng/ml tetracycline were used for induction, respectively, as depletion of Atlastins led to reduced expression levels of the reporters. For in vivo INM targeting using the RUSH system, expression of the LBR(1-245)-SBP-EGFP and Lap2β-SBP-EGFP reporters was induced with 5 ng/ml and 10 ng/ml tetracycline, respectively, 16 hr before the experiment. Cell lines used in this study were not further authenticated after obtaining from the sources. All cell lines were tested negative for mycoplasma using PCR-based testing. None of the cell lines used in this study were included in the list of commonly misidentified cell lines maintained by International Cell Line Authentication Committee.

## Protein expression and purification

For purification of dominant-negative Atlastin fragments (cytATLs), proteins were expressed in *E.coli* BLR(pREP4). Expression was induced by addition of 1 mM IPTG at 25°C for 4 hr. Cells were harvested and sonicated in lysis buffer (50 mM Tris pH 7.5, 700 mM NaCl, 3 mM MgCl$_2$ 5% w/v glycerol). The lysate was subjected to ultracentrifugation and the supernatant was passed over a Ni-NTA agarose column. After washing, proteins were eluted with lysis buffer containing 400 mM imidazole. The purified proteins were re-buffered into permeabilization buffer (20 mM Hepes pH 7.5, 110 mM potassium acetate, 5 mM magnesium acetate, 0.5 mM EGTA, 250 mM sucrose).

## RNA interference and transient transfections

siRNA transfection was performed in Opti-MEM medium using INTERFERin (Polyplus transfection), with a final siRNA concentration of 20 nM. The following siRNAs were used:

ATL2 (5'-TGGGAGATTGAAAGATATTGA-3'; Qiagen),
ATL3a (5'-CACGGGCATTGTAGCTTTGTA-3'; Microsynth),
ATL3b (5'-GGGCTACATCAGGTATTCTGGTCAA-3'; Qiagen),
ATL3c (5'-TACCGTATGTATTAAACCCAT-3'; Qiagen),
ATL3d (5'-CACGGGCATTGTAGCTTTGTA-3'; Qiagen),
ATL3e (5'-CAGGTTCATATCCAGAGGAAT-3'; Qiagen),
ATL3f (5'-CTCCTGTGGTTTCAAATTATT-3'; Qiagen).

Allstars (QIAGEN) was used as a negative control. RNAi was performed for 72 hr. Plasmid transfections were performed in Opti-MEM medium using the X-tremeGene transfection reagent (Roche). For microinjection experiments, U2OS cells were transiently transfected with 0.1 μg GFP-KDEL, 48 hr before the experiment. For overexpression of Atlastin dominant-negatives, DNA constructs were transfected 24 hr before the experiment. For analysis of ER exit, HeLa K cells were transiently transfected with 0.25 μg of the Str-Ii_VSVGwt-SBP-EGFP vector (*Boncompain et al., 2012*) 24 hr before the experiment.

## Generation of ATL3 KO cell lines using CRISPR/Cas9

ATL3 knockout (KO) cell lines were generated using the CRISPR/Cas9 system. CRISPR target sites in ATL3 were chosen within exon 2, which is found in all predicted splice variants of ATL3. Target sequences for guide RNAs (gRNAs) were predicted using a CRISPR design web tool (http://crispr.mit.edu). To clone the ATL3-Exon2 gRNA, the DNA oligonucleotides (5'-CACCGCATGTTGTCCCCTCAGCGAG-3') and (5'-AAACCTCGCTGAGGGGACAACATGC-3') were annealed and ligated into the pC2P vector, also encoding for hCas9 and a puromycin resistance cassette (kind gift from Dr. M. Bühler; Friedrich Miescher Institute for Biomedical Research, Basel). Cells were transfected with the gATL3Ex2-pC2P vector and selected with puromycin for two days. Cell colonies were picked, expanded and screened for *ATL3* mutations by PCR and immunoblotting. PCR products were sequenced and analyzed for insertions/deletions around the target sequence using the tide web tool (http://tide.nki.nl, [*Brinkman et al., 2014*]).

## RT-qPCR

Depletion of ATL3 relative to actin mRNA was analyzed by RTq-PCR using the following primers: ATL3 - forward (5'-TGCAGGTTGTTTTGGTTCAG-3'), reverse (5'-TTGAATGGCCACTTTCCTTC-3'); actin - forward (5'-TCCCTGGAGAAGAGCTACGA-3'), reverse (5'-AGCACTGTGTTGGCGTACAG-3'). RNA was extracted from control and siRNA treated HeLa cells using the RNeasy minikit (QIAGEN). Contaminating genomic DNA was digested with RNAse-free DNase I (Fermentas). 100 ng of DNase-treated RNA was then used per reverse transcription (RT) reaction. RNA was incubated with oligo dT18 primer (Fermentas), random hexamer primer (Fermentas), and RNase-free dNTPs at 65°C for 5 min. Reverse transcription was performed by adding SuperScript II Reverse Transcriptase (Invitrogen), 0.01 M DTT and RiboLock RNase inhibitor (Fermentas) followed by incubation at 42°C for 2 min, 25°C for 10 min, and 42°C for 1 hr. 2.5 ng cDNA was used for quantitative PCR. Primers were added together with LIghtCycler 480 SYBR Green I Master mix (Roche) to the template. The following amplification program was used: Initial denaturation at 95°C for 5 min followed by 40 cycles of 95°C for 10 s, 60°C for 10 s, and 72°C for 12 s. Results were analyzed using the ΔΔCt-method.

## In vitro INM targeting assays

In vitro INM targeting reactions were performed as described previously (*Ungricht et al., 2016*). In brief, reporter cells were semi-permeabilized with permeabilization buffer (PB: 20 mM Hepes pH 7.5, 110 mM potassium acetate, 5 mM magnesium acetate, 0.5 mM EGTA, 250 mM sucrose) containing 0.0025% digitonin for 10 min at 4°C, followed by three washes with PB for 2, 5, and 10 min, respectively. Targeting of reporter proteins towards the INM was initiated by proteolytic cleavage of their N-terminal 2xRFP moieties by addition of NusA-Tev protease to the semi-permeabilized cells for 10 min at room temperature. Targeting reactions were performed in presence of HeLa cell lysate and energy mix (10 mM creatine phosphate, 0.5 mM ATP, 0.5 mM GTP, 0.05 mg creatine kinase, 5 mM HEPES pH 7.5, and 12.5 mM sucrose), unless indicated otherwise. For competitive reactions, nucleotide analogs were added to a concentration of 0.3 mM. For energy depletion, 50 units of apyrase (New England Biolabs, Inc.) were added per reaction.

For experiments with dominant-negative Atlastins and Rab GDI, reporter cells were semi-permeabilized and pre-incubated for 20 min with 5 µM of dominant-negative Atlastins, 20 µM of Rab-GDI or both in PB at 37°C. Then, cells were exposed to NusA-Tev protease for 10 min. INM targeting reactions were carried out with HeLa cell lysate and energy mix in the presence of 5 µM dominant-negative Atlastins, 20 µM of Rab GDI or both. After 45 min, the coverslips were washed once in PB and immediately fixed with 4% paraformaldehyde. Nuclei were stained with DAPI. Imaging was performed at a Leica TCS SP2-AOBS microscope using the HCX Plan-Apochromat Lbd Bl 63X, NA 1.4, oil immersion lens.

Quantification of INM targeting levels was done as described (*Ungricht et al., 2016*). An ImageJ plugin was used to detect nuclear contours of DAPI-stained nuclei and the ER border by thresholding on the eGFP channel. The fluorescence intensity in a region defined as NE was integrated, comprising an area reaching 230 nm outwards from the nuclear contour towards the ER and 920 nm inwards into the nucleus. The fraction of total fluorescence intensity at the NE was calculated by dividing the fluorescence intensity at the NE by the sum of the integrated fluorescence intensities at the NE and the ER. Fluorescence intensities were normalized relative to the fraction at the NE before release by TEV cleavage (*Ungricht et al., 2015*). These normalized fluorescence intensities were then converted to the percentage of signal at NE.

## Analysis of INM targeting in living cells

For kinetic measurement of INM targeting in living cells, reporter cells were grown in Lab-Tek chambers. These were mounted in a chamber with 5% $CO_2$ at 37°C on an LSM-780-FCS microscope (Carl Zeiss) equipped with a 63X, 1.4 NA, oil DIC Plan-Apochromat immersion lens. Positions for imaging were marked and a pre-release image was acquired. Release of proteins for targeting to the INM was induced by addition of 250 µM biotin. Time-lapse imaging was performed for either 120 min or 90 min as indicated, acquiring images over z-stacks every 5 min. Integrated pixel intensities at the NE and in the ER were calculated using ImageJ. The fraction of the fluorescence intensity at the NE was calculated as described for the in vitro assay. Fluorescence intensities were normalized relative to the fraction at the NE before biotin addition (defined as 0.16 from an average of 59 cells) and converted to percent at the NE.

## Analysis of secretory trafficking in living cells

For kinetic measurement of secretory trafficking in living cells, HeLa cells were grown in Lap-Tek chambers and transfected with Str-Ii_VSVGwt-SBP-EGFP (*Boncompain et al., 2012*). These were mounted in a chamber with 5% $CO_2$ at 37°C on an Visitron spinning disk Nikon Eclipse T1 microscope with a 60X, 1.4 NA, CFI, Plan Apo λ Oil objective. Positions for imaging were marked and a pre-release image was acquired. Release of proteins for targeting to the Golgi complex was induced by addition of 250 µM biotin. Time-lapse imaging was performed for 15 min, acquiring images over z-stacks every 1 min. Integrated pixel intensities at the Golgi and in the whole cell were calculated using ImageJ. Fluorescence intensities were normalized relative to the fraction at the Golgi complex before biotin addition.

## FRAP and FLIP

FRAP and FLIP experiments were performed at an LSM 780-FCS microscope (Carl Zeiss) using a 63X, 1.4 NA, oil DIC Plan-Apochromat immersion lens as described previously (*Ungricht et al., 2015*). For FRAP experiments, reporter cells were seeded in 8-well Lab-Tek II chambers. Expression of the reporter proteins was induced for 16 hr. Then, cells were semi-permeabilized at 4°C, followed by addition of the indicated supplements and incubation at 37°C. To measure the diffusional mobility of the reporters in the ER, FRAP was performed on the uncleaved reporters. A $2 \times 10.4$ μm (area: 20.8 μm$^2$) region of interest (ROI) was defined. Three pre-bleach images were acquired, followed by bleaching with 100% laser power and 20 scanning iterations. The effective time for bleaching was 2.05 s. Pre- and post-bleach images were acquired every 4 s. Fluorescence intensities was also measured in equivalent areas inside and outside the cell for bleaching and background corrections. For quantitative analysis of FRAP experiments, fluorescence intensity values were imported to the easy-FRAP tool in MATLAB (*Rapsomaniki et al., 2012*). FRAP recovery curves were obtained using full-scale normalization. Mobile fractions and $t_{1/2}$ values were determined using a double exponential curve fit. Diffusion coefficients were calculated using the obtained $t_{1/2}$ values (*Kang et al., 2012*).

FLIP was performed on transiently transfected HeLa cells expressing GFP-KDEL (*Ungricht et al., 2015*). Two pre-bleach images were acquired, followed by repeated photo-bleaching of a spot (r = 3.16 μm, 50 iterations, 4.5 s). Fluorescence loss was measured within this spot, in a ring of 3.16 μm width surrounding the spot (defined as 'donut') and in a distant region on the opposite side of the nucleus. Fluorescence intensities within a neighboring cell and a background area were also measured to correct for bleaching while acquisition and background, respectively.

## Immunofluorescence assays

Cells were washed with PBS and fixed with 1% paraformaldehyde for 10 min. Then, cells were permeabilized with 0.2% Triton X-100 and blocked with 2% BSA in PBS. For immunostaining of POM121, cells were fixed and permeabilized using ice-cold methanol at −20°C for 5 min. After incubations with the primary antibody for 1 hr, cells were washed 3 times with 2% BSA in PBS and then incubated with the secondary antibody for 30 min. Cells were washed again with 2% BSA/PBS and PBS. Coverslips were mounted with Vectashield mounting medium (Vector Laboratories, Inc.). Imaging was performed at a Leica TCS SP2-AOBS microscope using a HCX Plan-Apochromat Lbd Bl 63X, NA 1.4 oil immersion lens.

## Microinjection and analysis of ER dynamics

U2OS cells were seeded on a glass bottom dish, well size 14 mm (Cellvis) and transfected with 0.1 μg of a vector encoding for EGFP-KDEL 48 hr before the experiment. Subsequently, a mixture of Alexa Fluor 647 conjugated dextran (0.5 mg/ml, MW 11 kDa) (Thermo Fischer Scientific) and GTP or GTPγS (10 mM) in permeabilization buffer was injected into the cytoplasm of cells using an Eppendorf FemtoJet and InjectMan Ni2 microinjection unit at a Leica DMI6000B inverted microscope equipped with a 20X, 0.4 NA, Ph1 HCX PlanFluotuar LD (long distance) objective. ER dynamics in microinjected cells was imaged for 1 min at a frame rate of 1/s using a Visitron spinning disk Nikon Eclipse T1 microscope with a 100X, 1.49 CFI, Apo TIRF objective. Image analysis was performed with ImageJ. Brightness and contrast were adjusted across the images. From each cell, 1–2 ROIs of 100 μm$^2$ in the peripheral ER were selected for manual analysis of successful or unsuccessful membrane attachments.

## Statistical analysis

Statistical significance was calculated using the unpaired t-test with Prism.

## Acknowledgements

We thank Drs. C. Blackstone (NINDS, Bethesda, USA), M. Bühler (FMI Basel, Switzerland; NCCR 'RNA and disease'), A. Helenius (ETH Zurich, Switzerland), F. Perez (Institut Curie, Paris, France) for providing reagents, Drs. C. Azzalin (IMM, Lisboa, Portugal), M.Beck (EMBL, Heidelberg, Germany), D. Gerlich (IMBA, Vienna, Austria) for providing cell lines, Dr. R. Klemm (University of Zurich, Switzerland) for helpful advice, K. Frischer-Ordu and Dr. S. Jonas for critical comments on the manuscript,

C. Ashiono for excellent technical assistance, and members of the ETH imaging facility ScopeM for continuous support. This work was funded by a European Research Council grant (NucEnv) to UK.

## Additional information

### Funding

| Funder | Grant reference number | Author |
|---|---|---|
| H2020 European Research Council | ERC Advanced Grant NucEnv | Sumit Pawar Ulrike Kutay |

The funders had no role in study design, data collection and interpretation, or the decision to submit the work for publication.

### Author contributions

Sumit Pawar, Conceptualization, Data curation, Formal analysis, Visualization, Writing—original draft; Rosemarie Ungricht, Conceptualization, Investigation, Visualization; Peter Tiefenboeck, Investigation, Methodology; Jean-Christophe Leroux, Supervision, Investigation, Methodology; Ulrike Kutay, Conceptualization, Supervision, Funding acquisition, Writing—original draft, Project administration

### Author ORCIDs

Sumit Pawar  https://orcid.org/0000-0003-3161-7524
Ulrike Kutay  http://orcid.org/0000-0002-8257-7465

### Decision letter and Author response

Decision letter https://doi.org/10.7554/eLife.28202.023
Author response https://doi.org/10.7554/eLife.28202.024

## Additional files

### Supplementary files

• Transparent reporting form
DOI: https://doi.org/10.7554/eLife.28202.021

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
