## [Decision Letter]

Thank you for submitting your article "Efficient protein targeting to the inner nuclear membrane requires Atlastin-dependent maintenance of ER topology" for consideration by *eLife*. Your article has been reviewed by three peer reviewers, and the evaluation has been overseen by Randy Schekman as the Reviewing and Senior Editor. The following individuals involved in review of your submission have agreed to reveal their identity: James A McNew (Reviewer #2) and Brian Burke (Reviewer #3).

The reviewers have discussed the reviews with one another and the Reviewing Editor has drafted this decision to help you prepare a revised submission.

Summary

How membrane proteins that reside in the inner nuclear membrane (INM) move to this location from their sites of synthesis in the endoplasmic reticulum (ER) is unclear, and various mechanisms have been suggested. The authors of this study have previously proposed a diffusion-retention-based mechanism of INM targeting that is energy-dependent (Ungricht et al., 2015). They also proposed that transport from the ER to the INM occurs through the pore membrane and is regulated by nuclear pore complex proteins. Importantly, they showed that these targeting events do not require RanGTP, and they conclude that the nature of this energy-dependent step remains to be determined.

In this manuscript, the authors further examine the functional basis for energy-dependent membrane protein targeting to the INM. To do this, they use a previously published reporter protein assay to assess the inducible movement of INM reporters from the ER to the INM in semi-permeabilized cells. They show using this assay that GTP is required for efficient movement of INM reporters to the INM. They further show that treatment with a nonhydrolyzable analog of GTP reduces the mobility of the reporters in the ER membrane and alters ER morphology. In addition, the authors show that conditions that inhibit the function of a member of a family of proteins termed atlastins delays recruitment of the INM reporter to the INM and alters the diffusion properties of the reporter within ER membranes. Based on these data and previously proposed functions of atlastins in homotypic ER membrane fusion, the authors conclude that atlastins contribute to INM protein targeting, in part, through their role in dictating ER topology.

This is a very solid piece of work. The experiments are well performed and the data are of excellent quality.

Essential revisions:

1) The authors need to clearly state that the requirement for GTP in INM protein targeting may not be limited to its use by atlastins. I think this is important as the role of NTPs in this process has been a debated point in the field, and more needs to be done to clarify this issue. Second, I am concerned that the effects of altering atlastin function are pleiotropic, and the consequences of atlastin loss of function to targeting of endogenous INM protein targeting is not clearly established. However, I don't have a seminal experiment to suggest. Perhaps the authors could at least state that more studies are needed to clearly define the function of atlastins in the targeting of endogenous INM proteins.

2) For the non-expert, it may be useful for the authors to make a clearer distinction between nuclear envelope accumulation and INM targeting. These terms seem to be used interchangeably.

3) Additionally, given that the majority of results in this work rely on nuclear envelope accumulation as an endpoint, it would be very useful if the authors demonstrated that NE accumulation faithfully recapitulated INM sorting in the context of atlastin 2 knockdown using the differential permabilization experiment with triton and digitonin. This may be partly the purpose of Figure 6, but it is unclear if the immunolabelling was done following Triton X-100 or digitonin permeabilization.

4) With respect to the Atlastin results, I have only a single query. INM protein localization appears insensitive to ATL3 knockdown but is inhibited by cytATL3 expression. My first thought was that this might simply reflect lower expression levels of ATL3 versus ATL2. However, export of VSV G protein appears to be inhibited by both ATL2 and ATL3 knockdown. Why might this be the case? Is it possible that accessing ER exit sites in the peripheral ER is more sensitive to even small changes in ER topology than accessing the ONM and associated ER cisternae? Perhaps the authors could address this in an amended discussion.

---

## [Author Response]

*Essential revisions:*

*1) The authors need to clearly state that the requirement for GTP in INM protein targeting may not be limited to its use by atlastins. I think this is important as the role of NTPs in this process has been a debated point in the field, and more needs to be done to clarify this issue.*

We agree that we can currently not fully exclude the involvement of additional GTPases in INM targeting. Therefore, we had already commented on this issue in the manuscript both in the Result and Discussion sections, and we would like to bring this to the notice of the reviewers.

Results:

‘This requirement of Atlastins, in turn, could explain the observed GTP-dependent nature of INM targeting (Figure 1). It does, however, not exclude the contribution of additional GTPases.’

Discussion:

‘Currently, we cannot exclude the involvement of additional GTPases in INM targeting. […] However, the respective mutants behaved as dominant-negatives, hampering the analysis.’

Further, we have devoted an entire section of the Discussion to the RanGTPase system and the differences between transport of soluble proteins and INM targeting of membrane proteins.

*Second, I am concerned that the effects of altering atlastin function are pleiotropic, and the consequences of atlastin loss of function to targeting of endogenous INM protein targeting is not clearly established. However, I don't have a seminal experiment to suggest. Perhaps the authors could at least state that more studies are needed to clearly define the function of atlastins in the targeting of endogenous INM proteins.*

Depletion of ATL2 or double depletion of both ATL2 and 3 perturbs ER topology to a large extent, as exemplified in Figure 4. Further, depletion of both atlastins leads to defects in the biogenesis of NPCs (Figure 6), which are the bottleneck for transport of proteins to the INM (Ungricht et al., 2015; Boni et al., 2015). To address the concern that the effects of altering Atlastin function are pleiotropic, we performed experiments in which we acutely inhibited Atlastin function by the use of dominant-negatives (Figure 5). In this experimental setting, we did not observe any effects on NPCs, as expected, but INM targeting was impaired. We have also analyzed the effects on targeting of endogenous INM proteins (Figure 6—figure supplement 2). Consistent with the notion that Atlastin depletion kinetically delays INM targeting but does not block it, one does not necessarily expect to see changes in the steadystate localization of INM proteins. In the manuscript, we point at the fact that ‘newly synthesized INM proteins present only a minor fraction of the total pool at a given time.’ Thus, it will be difficult to observe striking differences in the steady-state localization of endogenous INM proteins. And indeed, as we demonstrate (Figure 6—figure supplement 2), defects in INM targeting are only observed in post-mitotic cells in early G1, when a large pool of membrane proteins needs to be synchronously targeted back to the INM.

*2) For the non-expert, it may be useful for the authors to make a clearer distinction between nuclear envelope accumulation and INM targeting. These terms seem to be used interchangeably.*

Yes, we had used these terms interchangeably, as the NE accumulation seen in our assays majorly reflects INM localization as demonstrated previously (Ungricht et al., 2015). To avoid confusion, we now use the term NE accumulation when referring to the data derived from our measurements. Further, we have added an additional experiment using antibody accessibility upon differential detergent permeabilization to demonstrate INM localization of the LBR reporter in control cells and the targeting defects observed upon Atlastin depletions (Figure 4—figure supplement 3).

*3) Additionally, given that the majority of results in this work rely on nuclear envelope accumulation as an endpoint, it would be very useful if the authors demonstrated that NE accumulation faithfully recapitulated INM sorting in the context of atlastin 2 knockdown using the differential permabilization experiment with triton and digitonin. This may be partly the purpose of Figure 6, but it is unclear if the immunolabelling was done following Triton X-100 or digitonin permeabilization.*

As stated above, we now included this important control as new Figure 4—figure supplement 3 in the manuscript.

*4) With respect to the Atlastin results, I have only a single query. INM protein localization appears insensitive to ATL3 knockdown but is inhibited by cytATL3 expression. My first thought was that this might simply reflect lower expression levels of ATL3 versus ATL2.*

These are very valid comments. With respect to retardation of INM targeting upon cytATL3 addition, we reason that this may be due to the formation of mixed dimers between exogenously added cytATL3 and endogenous Atlastin 2 (as well as homodimers with ATL3), as has been previously observed for over-expressed proteins by Rismanchi et al., 2008. We had already stated this in our Discussion:

‘The potent impairment of INM targeting by cytATL3 may, however, suggest that Atlastins form mixed dimers under those conditions using excess recombinant protein, similar to what has been observed before (Rismanchi et al., 2008).”

The clear difference in phenotypic defects upon loss of ATL2 or 3 is indeed intriguing. Like the reviewers assumed, we also initially thought that this could be due to higher levels of ATL2 compared to ATL3 in HeLa cells. However, we have tried to estimate Atlastin levels by quantitative immunoblotting of HeLa cell extracts. These preliminary experiments suggested that ATL3 might be more abundant in HeLa cells than ATL2. As quantitative immunoblotting can sometimes lead to artifacts, we conclude that future work must use adequate assays like quantitative mass spectrometry to measure the relative abundance of Atlastins in different cell lines and tissues.

*However, export of VSV G protein appears to be inhibited by both ATL2 and ATL3 knockdown. Why might this be the case? Is it possible that accessing ER exit sites in the peripheral ER is more sensitive to even small changes in ER topology than accessing the ONM and associated ER cisternae? Perhaps the authors could address this in an amended Discussion.*

We would like to note that the defects in enrichment of the VSVG reporter at the Golgi upon loss of ATL3 are mild as compared to those observed upon loss of ATL2. Still, as noted by the reviewers, ATL3 depletion affects ER exit, and we currently do not have a molecular explanation for this observation. It could be that ER to Golgi transport is more sensitive to ATL3 depletion because ATL3 depletion specifically affects membrane organization of domains relevant for ER to Golgi transport. Interestingly, the Blackstone laboratory has observed differences in the localization of ATL2 and ATL3 (Rismanchi et al., 2008). We now comment on this issue in the Discussion, to read:

‘Intriguingly, ER exit of VSVG was not only perturbed upon depletion of ATL2 but also slightly affected by loss of ATL3. […] This may simply reflect ER escape of Atlastins accompanying the flux of membranes to the Golgi. Alternatively, it could also hint at a more specific role of Atlastins in maintaining the topology of the transitional ER.’